# Declining trend of smoking and smokeless tobacco in India: A decomposition analysis

**Supriya Lahoti**[1], **Priyanka Dixit**[2]*

**1** Master of Public Health (Health Policy, Economics and Finance), Tata Institute of Social Sciences, Mumbai, Maharashtra, India, **2** Centre for Health and Social Sciences, School of Health Systems Studies, Tata Institute of Social Sciences, Mumbai, Maharashtra, India

* priyanka.dixit@tiss.edu

**Data Availability Statement:** The data underlying this study is publicly available at: GATS-1 and 2 India Data https://www.who.int/tobacco/surveillance/survey/gats/gats_india_report.pdf?ua=1.

## Abstract

There has been a relative reduction of tobacco consumption between Global Adult Tobacco Survey-India (GATS-India) 2009–10 and GATS-India 2016–17. However, in terms of absolute numbers, India still has the highest number of tobacco consumers. Therefore, this paper aims to examine the socioeconomic correlates and delineate the factors contributing to a change in smoking and smokeless tobacco use from GATS (2009–10) to GATS (2016–17) in India. We used multivariable binary logistic regressions to examine the demographic and socioeconomic correlates of smoking and smokeless tobacco use for both the rounds of the survey. Further decomposition analysis has been applied to examine the specific contribution of factors in the decline of tobacco consumption over a period from 2009 to 2016. Results indicated that the propensity component was primarily responsible for major tobacco consumption decline (smoking- 41%, smokeless tobacco use- 81%). Most of the decrease in propensity to smoke has been explained by residential type and occupation of the respondent. Age of the respondent contribute significantly in reducing the prevalence of smokeless tobacco consumption during the seven-year period, regardless of change in the composition of population. To achieve the National Health Policy, 2017 aim of reducing tobacco use up to 15% by 2020 and up to 30% by 2025, targeted policies and interventions addressing the inequalities identified in this study, must be developed and implemented.

## Introduction

Tobacco consumption globally is one of the leading causes of potentially preventable morbidity and mortality [1]. Tobacco stands as the leading cause for non-communicable disease (NCD) globally and mortality due to NCDs accounts to about 63% [2]. Globally, 80% of the tobacco-related deaths occur in the Low and Middle Income Countries (LMIC) [3]. According to World Health Organization's (WHO) estimation, deaths due to tobacco-related diseases will rise from 1.4% in 1990 to 13.3% in 2020 [4]. Projected tobacco-associated mortality in India is estimated to be 1.5 million by 2020 [5]. Tobacco use is broadly categorised into two main forms, smoking and smokeless. India's tobacco consumption situation is very complex, with not only a variety of smoking forms but also an array of smokeless tobacco products.

**Funding:** The author(s) received no specific funding for this work.

**Competing interests:** The authors have declared that no competing interests exist.

Further, the choice of product and form of tobacco consumption are influenced by the interplay of various demographic, social, economic and cultural practices. In India predominantly, smoking is by way of *bidis* (tobacco hand-rolled and wrapped in dried leaves of specific trees) and cigarettes. Smokeless tobacco use consists of *khaini*, chewing *pan* (mixture of lime, areca nut pieces, tobacco and spices wrapped in betel leaf), *gutkha* or *pan masala* (powder mixture of scented tobacco, lime and areca nut), and *mishri* (a kind of toothpaste used for rubbing on gums) [6]. It has been documented that tobacco use in any form can result in serious health consequences [7].

A trend analysis of tobacco use in India, using nationally representative surveys documented an increase in the prevalence of any smokeless tobacco use from 15% in 1987 to 23.4% in 2005 while slight decline in any smoked tobacco from 19.8% to 18.3% in the same period [8]. Recent data in India shows that from Global Adult Tobacco Survey (2009–2010) to Global Adult Tobacco Survey (2016–2017), there has been a 4.5% decline, in prevalence of smokeless tobacco use from 25.9% to 21.4% and a 3.3% decline in smoking, from 14.0% to 10.7% [9]. The effect of socio-economic status on the prevalence of tobacco use has been documented well in literature [10–13]. Previous studies have shown that community level (regions [10], place of residence [12]), household level (wealth status [14]) and individual level factors (age [12, 15], gender [15]) are associated with smoking and smokeless tobacco use in India.

In analytical terms, there are two main mechanisms through which the aggregate level of tobacco consumption can change over two points of time. One potential source of change is a shift in the proportion of population from one social group to another, which typically has low rates of any particular form of tobacco consumption (e.g. increase in the proportion of educated population). Aggregate change may also result from an increase in the likelihood of decreasing tobacco consumption among all subgroups or among those subgroups that had higher rates of tobacco consumption at an earlier time (reflecting diffusion processes, convergence of tobacco consumption behaviour among social strata and a conceivably deliberate programme targeting less favoured groups).

This study decompose overall change into these two underlying forces: compositional change and the rate of change. Decomposition analysis provides relevant insights into the causal mechanisms that underlie the observed trend of decline in tobacco consumption. Previous studies were mostly limited to examining tobacco consumption trends and its correlates in India [10–15]. To the best of our knowledge there are no studies examining the specific contribution of factors in tobacco consumption decline over a period from 2009 to 2016. This paper, for the first time employs two nationally representative comparable tobacco consumption related data sets and desegregates the change in tobacco consumption in the smoking and smokeless forms in the different socio-economic and demographic sub-groups, allowing us to identify which factors contributed to the documented change in tobacco consumption in the inter-survey period. This analysis is potentially useful from a policy perspective given that it provides policy makers in the country with insights to address inequalities in tobacco consumption in pursuit of the 2030 agenda of Sustainable Development Goal of poverty reduction and good health.

## Materials and methods

### Data sources: Sample size and design

The data used for the present study has been gleaned from the Global Adult Tobacco Survey 2009–2010 (hereafter referred to as GATS-1) [16] & Global Adult Tobacco Survey 2016–2017 (hereafter referred to as GATS-2) [9] conducted in India. The Ministry of Health & Family Welfare (MoHFW), Government of India designated the International Institute for Population

Sciences (IIPS), Mumbai and the Tata Institute of Social Sciences (TISS), Mumbai as the nodal implementing agency for GATS-1 and GATS-2 respectively. Data sets of both surveys are sufficiently similar to allow the construction of important predictors of different forms of tobacco use. Moreover, the interviewing procedures and sampling design were similar in both surveys and the design was such that it represents Indian residents. The GATS is a nationally representative, multi-stage, geographically clustered sample of households that covered men and women above 15 years of age in India's 30 states (29 states in GATS-1) and two Union Territories (UTs). Multistage sampling procedure was adopted independently in each state, and within the states, independently in urban and rural areas to select the sample. In urban areas, a three-stage sampling process was adopted. At the first stage, the list of all the wards from all cities and towns of the state/ UT constituted the urban sampling frame, from which a required sample of wards, i.e., primary sampling units (PSUs) was selected using probability proportional to size (PPS) sampling. At the second stage, a list of all census enumeration blocks (CEBs) in each selected ward constituted the sampling frame from which one CEB was selected by PPS from each ward. At the third stage, a list of all residential households in each selected CEB constituted the sampling frame, from which a sample of required number of households was selected.

In rural areas, a two-stage sampling process was adopted. At the first stage of sampling, PSUs (village) were selected using the PPS sampling method. At the second stage, a list of all residential households in each selected village constituted the sampling frame, from which a sample of the required number of households was selected. From each eligible household, one respondent was selected. More details about sampling design, training of the survey team, and survey management are separately documented in GATS-1 and GATS-2 published report [9, 16].

After excluding the incomplete cases, the total sample size was reduced to 69,296 and 74,037 residents aged 15 years or above in GATS-1 and GATS-2 respectively. The overall response rate calculated as the product of the response rates at the household and person-level was 91.8 percent and 92.9 percent for GATS-1 and GATS-2 respectively.

The main objective of the GATS survey was to collect reliable information on tobacco use and tobacco control indicators in order to develop an understanding of the effectiveness of tobacco control measures undertaken during the inter-survey period. We used data from the two rounds of GATS to provide national-level estimates of different types of tobacco users and socio-economic and demographic correlates of two types of tobacco use (smoking and smokeless tobacco). GATS provides information on respondents' background characteristics, tobacco use (smoking and smokeless), cessation, second hand smoke exposure, economics, media, and knowledge, attitudes and perceptions towards tobacco use.

### Independent variables

Many factors have significant effects on tobacco consumption. Based on available literature, relevant variables have been included in the model to decompose the change in the smoking and smokeless tobacco consumption separately. We broadly categorized these variables as community, household and individual level.

The variables included under the community category were Regions (North, Central, East, Northeast, West and South) and residence (urban/rural). Based on geographical location and cultural factors, India was divided into six regions: North (Jammu & Kashmir, Himachal Pradesh, Punjab, Chandigarh, Uttarakhand, Haryana and Delhi); Central region (Rajasthan, Uttar Pradesh, Chhattisgarh and Madhya Pradesh); East (West Bengal, Jharkhand, Odisha and Bihar); North-east (Sikkim, Arunachal Pradesh, Nagaland, Manipur, Mizoram, Tripura,

Meghalaya and Assam); West (Gujarat, Maharashtra and Goa) and South (Andhra Pradesh, Telangana, Karnataka, Kerala, Tamil Nadu and Puducherry). The new state of Telangana was created from Andhra Pradesh in the year 2014 and is included in southern region in the GATS-2 data.

The variables included under household category were wealth index quintiles (poorest, second, middle, fourth and richest), caste (Scheduled Castes [SCs] and Scheduled Tribes [STs], Other Backward Classes [OBCs], and Others [others caste includes non-SC, non-ST and non-OBC]) and religion (Hindu, Muslim and Others [Others religion includes non-Hindu and non-Muslims]).

The household wealth index was estimated using an asset index. The asset index was constructed based on household assets and possession of household consumer items using Principal Component Analysis technique. Based on time relevance, 10 and 14 household assets were included in GATS-1 and GATS-2 respectively to create the wealth index in the respective time period. Using rank methods, households were classified by wealth quintiles.

The variables included under individual category were age in completed years (15-24/25-44/45-64 and 65 and above), sex (male/female), level of education (no education, primary, secondary and higher), and type of occupation (government and non-government, self-employed, student, homemaker and retired/unemployed). Further, individual knowledge and perception variables included were knowledge that exposure to smoking causes serious illness, stroke, heart attack, lung cancer and chronic cough/ TB (Yes/No) and knowledge that smokeless tobacco causes serious illness, oral cancer and dental disease (Yes/No).

In multivariable regression and decomposition analysis, we have only included common variables available in GATS-1 and GATS-2, i.e. regions, type of residence, wealth index quintiles, age group, gender, education, occupation, knowledge that smoking causes serious illness, stroke, heart attack, lung cancer and knowledge that smokeless tobacco causes serious illness.

## Outcome variables

The different forms of smoking tobacco included were *bidis*, cigarettes, cigars, cheroots, rolled cigarettes, tobacco rolled in maize leaf and newspaper, *hukkah*, pipes, *chillum* and *chutta*. The different forms of smokeless tobacco included tobacco leaf, betel quid with tobacco, *sada/surti*, *khaini* or tobacco lime mixture, *gutkha*, *paan masala* with *zarda*, *mawa*, *gul*, *gudaku*, *mishri*. The main dependant variable in the analysis is the tobacco use categorized into two types, namely, smoking tobacco and smokeless tobacco use.

Tobacco consumption has been divided into two categories:

1. *Smoking*: all respondents who smoked tobacco ('daily' and 'less than daily') were coded as "1" whereas those who did not smoke ('never' and 'former') were coded as "0".

2. *Smokeless*: includes all respondents who consumed smokeless tobacco ('daily' and 'less than daily') were coded as "1" whereas those who did not use smokeless tobacco ('never' and 'former') were coded as "0".

## Statistical analysis

Bivariate analysis was used to estimate the prevalence of smoking and smokeless tobacco use in association with selected background variables described in the earlier section of GATS-1 and 2 data. The prevalence is presented in the form of percentages. The percent relative change was calculated using the formula (Prevalence in GATS-2)-(Prevalence in GATS-1)/ (Prevalence in GATS-1). The proportion of smokers and smokeless tobacco users was calculated

using Univariate analysis. Multivariable binary logistic regression is used to investigate and assess the adjusted associations of socioeconomic, demographic and knowledge correlates of tobacco consumption in the smoking and smokeless forms in India. In the multivariable logistic regression we have adjusted the impact of clustering and stratification.

Finally, decomposition analysis has been used to examine the role of various factors in declining tobacco consumption during the seven-year period. The decomposition technique adopted in this paper is a well-established demographic technique built upon Kitawaga' (1955) classical work on rate standardization [17]. This procedure yields three components: rates, composition and interaction. Rate change stands for the change in the likelihood of smoking and smokeless forms of tobacco consumption by the different social, economic and demographic subgroups of population as expressed by the β coefficients and constant terms of the binary regression, regardless of the change in the composition component. Compositional change refers to the structural changes in the population such as change in population literacy in the two time periods or a part of the overall change, which is ascribed to changes in the means of the covariates, keeping rate as a constant as it was in GATS-1. Interaction reflects the contribution of the change in smoking and smokeless tobacco use as a result of the interplay between compositional change and propensity to use tobacco by different socio-economic and demographic subgroups. With the help of this method, we have tried to determine the net contribution of each of the selected covariates to smoking and smokeless tobacco decline.

## Model

The model takes the form

$$\ln[P_i \div (1 - P_i)] = \sum \beta_i X_i$$

Where $\ln[P_i \div (1 - P)_i]$ are the log odds of tobacco consumption, $X_i$ is a vector of explanatory variables, and $\beta_i$ is a vector of regression coefficients. The decomposition procedure (smoking and smokeless tobacco use) applied in this study is based on the logit models (smoking and smokeless tobacco use) estimated for the two surveys.

The difference

$$ln[P_i \div (1 - P_i))_{(GATS-2)} - ln[P_i \div (1 - P_i)]_{(GATS-1)}$$

is decomposed using the following equation (which considered GATS-1 as the base period)

$$\log(GATS - 2) - \log(GATS - 1) = (\beta_{0(II)} - \beta_{0(I)})$$
$$+ \sum P_{ij(I)}(\beta_{ij(II)} - \beta_{ij(I)})$$
$$+ \sum \beta_{ij(I)}(P_{ij(II)} - P_{ij(I)})$$
$$+ \Sigma(P_{ij(II)} - P_{ij(I)})(\beta_{ij(II)} - \beta_{ij(I)})$$

Where $P_{ij(II)}$ = Proportion of the $j^{th}$ category of the $i^{th}$ covariate in GATS- 2;
$P_{ij(I)}$ = Proportion of the $j^{th}$ category of the $i^{th}$ covariate in GATS- 1;
$\beta_{0(II)}$ = Regression constant in GATS- 2;
$\beta_{0(I)}$ = Regression constant in GATS- 1;
$\beta_{ij(II)}$ = Coefficient for the jth category of the ith covariate in GATS- 2;
$\beta_{ij(I)}$ = Coefficient for the jth category of the ith covariate in GATS- 1;
$I$ denotes GATS- 1 and II denotes GATS- 2.

STATA command svyset gatscluster [pweight = gatsweight], strata (gatsstrata) has been used to adjust complex analysis which includes adjustment of clustering and stratum effect. The analysis of the data has been carried out after assigning survey weights that is available in the GATS-1 and GATS-2 datasets. While generating all the tables of this paper, each record (individual case) was multiplied by survey weight. These weights were estimated for adjustment of 1) unequal probability of selection, 2) differential response rates across states and male/ female in rural/ urban areas within the states and 3) differences in the distribution of survey population and actual population (projected as on survey period) of each state by rural/ urban areas and by sex and broad age-group. In other words, the weights were the adjustment within each individual state and across the states.

Further details of the weighting procedure are provided in section A 4, on GATS-1 report [16]. Data were analysed using IBM SPSS Statistics Version 22 (Armonk, New York, USA) and STATA Version 14.

## Results

### Prevalence of tobacco use by background characteristics

Table 1 presents the prevalence of tobacco in the form of smoking and smokeless tobacco use in India by different socioeconomic and demographic characteristics in GATS-1 and 2. The prevalence varies considerably by regions, rural-urban residence, age groups, gender, economic status of household and across some knowledge parameters. The prevalence of smoking was higher in the northern region of the country in GATS-1 (11.7%) and GATS-2 (12.0%), those living in the rural areas, from poor households, among males and illiterate, as compared to the other regions, urban residents, from the rich household and among literates. Smokeless tobacco use is highest in the north-eastern region and in rural areas of the country. Smokeless tobacco use shows an increase in the north-eastern region from GATS-1 to GATS-2 (24.9% to 32.6%). The rich poor differences were most visible among smokeless tobacco users. In both the rounds of the survey, minimum decline has been observed among the self-employed individuals in smoking (GATS-1:13.7%, GATS-2:12.3%) and smokeless tobacco use (GATS-1:26.9%, GATS-2:26.7%). Prevalence varies considerably by the knowledge of smoking association with lung cancer, and smokeless tobacco health hazards than the other two diseases.

Among smokers, prevalence of smoking tobacco varies considerably by the knowledge of smoking associated with lung cancer than the other three diseases in GATS-1. In GATS-2, prevalence of smokeless tobacco use among the users varies considerably by the knowledge of smokeless tobacco consumption associated with oral cancer than the other diseases.

### Multivariable binary logistic regression for smokers

Table 2 depicts the results of the binary logistic regression model to enhance the understanding of the role of different covariates (whether their effects have remained constant or not on smoking form of tobacco use in the two survey periods) of smoking in India. The proportion (P), β coefficients and adjusted odds ratios (exponential β) along with a 95 per cent confidence interval (CI) estimate of smokers according to the various categories of a variable compared to the reference category is presented. The last column of this table presents the difference in the regression coefficients during the seven-year period.

The coefficients in all the regions except the north east is found to be negative in both the rounds of the survey compared to the north region, indicating these regions had lower smoking levels as compared to North India. The eastern region followed by the central region has attained a higher rate of smoking decline as opposed to other regions. The rate of smoking has increased in rural areas during the inter-survey period.

**Table 1. Prevalence of smoking and smokeless use of tobacco according to selected background characteristics in India, GATS-2009-10 and GATS-2016-17.**

| Explanatory variables | Smoker | | | | Relative Change | Smokeless tobacco user | | | | Relative Change |
|---|---|---|---|---|---|---|---|---|---|---|
| | GATS-1 | | GATS-2 | | | GATS-1 | | GATS-2 | | |
| | n | % | n | % | | n | % | n | % | |
| **Region** | | | | | | | | | | |
| North | 4816224 | 11.7% | 9748877 | 12.0% | 2.9% | 2066425 | 5.0% | 4217336 | 5.2% | 3.8% |
| Central | 22940948 | 8.9% | 18407041 | 6.8% | -23.5% | 58501637 | 22.6% | 57719118 | 21.3% | -5.9% |
| East | 13047984 | 7.8% | 13260898 | 6.6% | -15.6% | 49961712 | 29.8% | 46656444 | 23.1% | -22.4% |
| Northeast | 2722448 | 9.5% | 3095637 | 9.0% | -5.7% | 7131380 | 24.9% | 11269000 | 32.6% | 31.1% |
| West | 6120010 | 5.2% | 4877760 | 3.5% | -32.6% | 26554721 | 22.4% | 29114576 | 20.8% | -7.3% |
| South | 19258634 | 10.7% | 18033548 | 8.9% | -16.8% | 19467462 | 10.8% | 18313850 | 9.0% | -16.4% |
| **Type of Residence** | | | | | | | | | | |
| Urban | 17790277 | 7.7% | 19123577 | 5.9% | -22.3% | 32737232 | 14.1% | 41608645 | 12.9% | -8.1% |
| Rural | 51115971 | 9.1% | 48300183 | 7.9% | -12.9% | 130946105 | 23.3% | 125681680 | 20.6% | -11.5% |
| **Wealth index** | | | | | | | | | | |
| Poorest | 12214464 | 9.1% | 13462085 | 8.3% | -7.9% | 41486565 | 30.8% | 45601811 | 28.0% | -9.0% |
| Poorer | 17626338 | 9.6% | 17112055 | 8.1% | -15.3% | 49823066 | 27.1% | 48770798 | 23.2% | -14.6% |
| Middle | 14224853 | 9.0% | 12877599 | 7.5% | -16.2% | 33202198 | 20.9% | 31043159 | 18.1% | -13.5% |
| Fourth | 13898419 | 8.6% | 14128400 | 7.0% | -19.3% | 24429997 | 15.2% | 28404285 | 14.0% | -7.7% |
| Richest | 10942173 | 6.9% | 9843622 | 5.3% | -24.0% | 14741510 | 9.4% | 13470271 | 7.2% | -22.8% |
| **Religion** | | | | | | | | | | |
| Hindu | NA | NA | 53155306 | 7.1% | NA | NA | NA | 136155164 | 18.2% | NA |
| Muslim | NA | NA | 11247299 | 8.5% | NA | NA | NA | 24229108 | 18.3% | NA |
| Others | NA | NA | 3021156 | 6.0% | NA | NA | NA | 6906053 | 13.6% | NA |
| **Caste** | | | | | | | | | | |
| SC & ST | NA | NA | 21430548 | 8.2% | NA | NA | NA | 59836487 | 23.0% | NA |
| OBC | NA | NA | 27960139 | 6.6% | NA | NA | NA | 69884502 | 16.6% | NA |
| Others | NA | NA | 18033073 | 7.2% | NA | NA | NA | 37569336 | 15.0% | NA |
| **Age groups** | | | | | | | | | | |
| 15–24 | 5354715 | 2.3% | 6573797 | 2.6% | 15.6% | 30729565 | 13.1% | 28291836 | 11.4% | -13.3% |
| 25–44 | 28543798 | 8.5% | 32417931 | 7.8% | -8.3% | 74906497 | 22.4% | 80752666 | 19.5% | -13.0% |
| 45–64 | 27458653 | 15.9% | 20808523 | 10.3% | -35.4% | 43245141 | 25.0% | 42933878 | 21.2% | -15.4% |
| 65 and above | 7549081 | 14.0% | 7623509 | 11.5% | -18.0% | 14802134 | 27.5% | 15311945 | 23.1% | -16.0% |
| **Gender** | | | | | | | | | | |
| Male | 61729247 | 15.0% | 60797567 | 12.8% | -15.0% | 97102345 | 23.6% | 111395181 | 23.4% | -1.0% |
| Female | 7177000 | 1.9% | 6626193 | 1.5% | -22.2% | 66580991 | 17.3% | 55895144 | 12.3% | -29.2% |
| **Education** | | | | | | | | | | |
| No formal schooling | 27191924 | 11.0% | 24663234 | 10.0% | -8.6% | 68213879 | 27.5% | 60628470 | 24.6% | -10.4% |
| Upto Primary | 20221853 | 10.6% | 18529170 | 9.7% | -8.9% | 45026561 | 23.7% | 44551669 | 23.3% | -1.6% |
| Upto Secondary | 14249400 | 6.3% | 16731392 | 5.8% | -7.3% | 36573764 | 16.1% | 45880945 | 15.9% | -1.0% |
| Above Secondary | 7243070 | 5.6% | 7477931 | 3.6% | -35.1% | 13869133 | 10.7% | 16087347 | 7.8% | -27.1% |
| **Occupation** | | | | | | | | | | |
| Government and non-government employee | 24284198 | 12.6% | 8169490 | 7.9% | -37.1% | 46592501 | 24.2% | 18266386 | 17.8% | -26.7% |
| Self employed | 31129017 | 13.7% | 46461593 | 12.3% | -10.5% | 61007114 | 26.9% | 101152182 | 26.7% | -0.5% |
| Student | 1089583 | 1.2% | 1136483 | 1.0% | -16.3% | 4827254 | 5.4% | 2928173 | 2.6% | -51.4% |
| Homemaker | 6350321 | 2.6% | 4732720 | 1.7% | -35.4% | 39861046 | 16.4% | 31840307 | 11.4% | -30.8% |
| Retired or unemployed | 5947826 | 13.7% | 6871052 | 11.6% | -15.3% | 11102842 | 25.6% | 13090757 | 22.1% | -13.6% |
| **Marital status** | | | | | | | | | | |
| Single | NA | NA | 5720679 | 2.7% | NA | NA | NA | 18605597 | 8.7% | NA |

*(Continued)*

**Table 1.** (Continued)

| Explanatory variables | Smoker | | | | Relative Change | Smokeless tobacco user | | | | Relative Change |
|---|---|---|---|---|---|---|---|---|---|---|
| | GATS-1 | | GATS-2 | | | GATS-1 | | GATS-2 | | |
| | n | % | n | % | | n | % | n | % | |
| Married | NA | NA | 57698130 | 8.8% | NA | NA | NA | 131773056 | 20.2% | NA |
| Separated | NA | NA | 223404 | 5.5% | NA | NA | NA | 877563 | 21.8% | NA |
| Divorced | NA | NA | 237826 | 6.3% | NA | NA | NA | 886952 | 23.4% | NA |
| Widowed | NA | NA | 3543721 | 6.3% | NA | NA | NA | 15121905 | 26.9% | NA |
| **Smoking causes serious illness** | | | | | | | | | | |
| Yes | 60124053 | 8.4% | 61907381 | 7.2% | -14.4% | 143116538 | 20.0% | 151261370 | 17.6% | -12.1% |
| No | 8682187 | 11.1% | 4493709 | 8.3% | -25.8% | 20269646 | 26.0% | 12555812 | 23.1% | -11.3% |
| **Smoking causes stroke** | | | | | | | | | | |
| Yes | 29466415 | 7.5% | 43758516 | 7.1% | -5.1% | 72892456 | 18.6% | 100549553 | 16.4% | -11.8% |
| No | 14224129 | 10.4% | 14446383 | 8.0% | -23.6% | 27318348 | 20.0% | 39957765 | 22.0% | 10.1% |
| **Smoking causes heart attack** | | | | | | | | | | |
| Yes | 39308632 | 7.7% | 51237641 | 7.2% | -7.4% | 92929275 | 18.3% | 117921732 | 16.5% | -9.9% |
| No | 9950110 | 10.9% | 10209391 | 8.1% | -25.9% | 21188958 | 23.2% | 29848852 | 23.6% | 1.8% |
| **Smoking causes lung cancer** | | | | | | | | | | |
| Yes | 54976269 | 8.2% | 61989668 | 7.1% | -12.8% | 132428115 | 19.6% | 152557315 | 17.5% | -10.9% |
| No | 3513766 | 17.1% | 3207903 | 9.0% | -47.4% | 4608607 | 22.4% | 9276323 | 26.0% | 16.0% |
| **Smoking causes chronic cough or TB** | | | | | | | | | | |
| Yes | NA | NA | 62410636 | 7.3% | NA | NA | NA | 150005144 | 17.4% | NA |
| No | NA | NA | 3150129 | 7.4% | NA | NA | NA | 10509832 | 24.8% | NA |
| **Smokeless tobacco use causes serious illness** | | | | | | | | | | |
| Yes | 57437868 | 8.1% | 63700270 | 7.1% | -12.3% | 141578562 | 20.1% | 156848841 | 17.6% | -12.4% |
| No | 5263821 | 14.2% | 2381842 | 9.2% | -35.7% | 8912298 | 24.1% | 7070705 | 27.2% | 12.8% |
| **Smokeless tobacco causes oral cancer** | | | | | | | | | | |
| Yes | NA | NA | 62831242 | 7.1% | NA | NA | NA | 153956683 | 17.5% | NA |
| No | NA | NA | 2575714 | 8.5% | NA | NA | NA | 8597977 | 28.4% | NA |
| **Smokeless tobacco causes dental disease** | | | | | | | | | | |
| Yes | NA | NA | 60459366 | 7.1% | NA | NA | NA | 148219859 | 17.5% | NA |
| No | NA | NA | 4596711 | 8.3% | NA | NA | NA | 13773748 | 24.9% | NA |

In the year 2009–10, the rate of smoking was higher in the poorer categories of wealth index quintiles (WIQ) after adjusting the other covariates in the model. In the year 2016–17, the middle, richer and richest population had lower levels of smoking as compared to their poorest counterparts. Although the direction of the coefficients for the above mentioned groups has become negative in the year 2016–17, the richest section has accomplished relatively higher gain in declining smoking levels. Women have lower levels of smoking as compared to men in both the rounds of the survey.

Education shows the expected inverse relationship with smoking. As the education level increases, the level of smoking declines. The difference in the rate of smoking between illiterate and educated categories becomes more prominent during the inter-survey period. It reflects that the effect of education on smoking had become stronger in 2016–17 than in 2009–10.

The table shows that in the first round of the survey, as compared to government and non-government employees, students and homemakers have lower levels of smoking. In the second round, the rate of smoking was higher in the self-employed and students' group. Students show an increase in the rate of smoking during the inter-survey period.

**Table 2. Multivariable binary logistic regression for smokers in India, GATS-2009-10 and GATS-16-17.**

| Explanatory variables | GATS-1 | | | | | GATS-2 | | | | | Change |
| --- | --- | --- | --- | --- | --- | --- | --- | --- | --- | --- | --- |
| | P | β | AOR | 95% C.I. of AOR | | P | β | AOR | 95% C.I. of AOR | | β gats-1 & β gats-2 |
| | | | | Lower | Upper | | | | Lower | Upper | |
| **Region** | | | | | | | | | | | |
| (North)[①] | | | 1 | | | | | 1 | | | |
| Central | 0.333 | -0.118 | 0.889 | 0.801 | 0.985 | 0.273 | -0.739 | 0.478 | 0.435 | 0.524 | -0.621 |
| East | 0.189 | -0.130 | 0.878 | 0.791 | 0.975 | 0.197 | -0.892 | 0.410 | 0.371 | 0.453 | -0.762 |
| Northeast | 0.04 | 0.573 | 1.773 | 1.629 | 1.930 | 0.046 | 0.494 | 1.638 | 1.512 | 1.775 | -0.079 |
| West | 0.089 | -1.103 | 0.332 | 0.293 | 0.376 | 0.072 | -1.448 | 0.235 | 0.207 | 0.267 | -0.346 |
| South | 0.279 | -0.369 | 0.691 | 0.627 | 0.763 | 0.267 | -0.644 | 0.525 | 0.481 | 0.574 | -0.275 |
| **Type of Residence** | | | | | | | | | | | |
| (Urban)[①] | | | 1 | | | | | 1 | | | |
| Rural | 0.742 | 0.019 | 1.019 | 0.954 | 1.088 | 0.716 | 0.090 | 1.094 | 1.026 | 1.167 | 0.071 |
| **Wealth index quintiles** | | | | | | | | | | | |
| (Poorest)[①] | | | 1 | | | | | 1 | | | |
| Poorer | 0.256 | 0.216 | 1.242 | 1.110 | 1.390 | 0.254 | -0.077 | 0.926 | 0.852 | 1.005 | -0.294 |
| Middle | 0.206 | 0.136 | 1.145 | 1.020 | 1.285 | 0.191 | -0.145 | 0.865 | 0.790 | 0.947 | -0.280 |
| Fourth | 0.202 | 0.066 | 1.069 | 0.950 | 1.202 | 0.21 | -0.251 | 0.778 | 0.708 | 0.856 | -0.317 |
| Richest | 0.159 | -0.065 | 0.937 | 0.827 | 1.063 | 0.146 | -0.453 | 0.636 | 0.571 | 0.707 | -0.388 |
| **Age groups** | | | | | | | | | | | |
| (15–24)[①] | | | 1 | | | | | 1 | | | |
| 25–44 | 0.414 | 0.787 | 2.198 | 1.967 | 2.456 | 0.481 | 0.709 | 2.033 | 1.806 | 2.288 | -0.078 |
| 45–64 | 0.398 | 1.066 | 2.904 | 2.580 | 3.268 | 0.309 | 1.028 | 2.795 | 2.472 | 3.161 | -0.038 |
| 65 and above | 0.11 | 0.926 | 2.525 | 2.164 | 2.945 | 0.113 | 0.783 | 2.189 | 1.908 | 2.511 | -0.143 |
| **Gender** | | | | | | | | | | | |
| (Male)[①] | | | 1 | | | | | 1 | | | |
| Female | 0.104 | -2.631 | 0.072 | 0.064 | 0.081 | 0.098 | -2.768 | 0.063 | 0.056 | 0.070 | -0.137 |
| **Education** | | | | | | | | | | | |
| (No formal schooling)[①] | | | 1 | | | | | 1 | | | |
| Upto Primary | 0.293 | -0.254 | 0.776 | 0.712 | 0.845 | 0.275 | -0.332 | 0.717 | 0.664 | 0.775 | -0.078 |
| Upto Secondary | 0.207 | -0.596 | 0.551 | 0.504 | 0.603 | 0.248 | -0.710 | 0.491 | 0.453 | 0.533 | -0.114 |
| Above Secondary | 0.105 | -0.915 | 0.400 | 0.360 | 0.445 | 0.111 | -1.215 | 0.297 | 0.267 | 0.329 | -0.300 |
| **Occupation** | | | | | | | | | | | |
| (Government and non-government employee)[①] | | | 1 | | | | | 1 | | | |
| Self employed | 0.452 | -0.035 | 0.966 | 0.903 | 1.033 | 0.69 | 0.236 | 1.266 | 1.166 | 1.374 | 0.271 |
| Student | 0.016 | -0.867 | 0.420 | 0.349 | 0.506 | 0.017 | -0.859 | 0.424 | 0.348 | 0.516 | 0.008 |
| Homemaker | 0.092 | -0.314 | 0.731 | 0.636 | 0.840 | 0.07 | -0.020 | 0.980 | 0.849 | 1.132 | 0.294 |
| Retired or unemployed | 0.086 | -0.092 | 0.912 | 0.808 | 1.029 | 0.102 | 0.002 | 1.002 | 0.888 | 1.129 | 0.094 |
| **Smoking causes serious illness** | | | | | | | | | | | |
| (Yes)[①] | | | 1 | | | | | 1 | | | |
| No | 0.126 | -0.152 | 0.859 | 0.603 | 1.223 | 0.068 | -0.102 | 0.903 | 0.806 | 1.011 | 0.050 |
| **Smoking causes stroke** | | | | | | | | | | | |
| (Yes)[①] | | | 1 | | | | | 1 | | | |
| No | 0.326 | 0.165 | 1.179 | 1.090 | 1.276 | 0.248 | 0.176 | 1.193 | 1.101 | 1.292 | 0.011 |
| **Smoking causes heart attack** | | | | | | | | | | | |
| (Yes)[①] | | | 1 | | | | | 1 | | | |
| No | 0.202 | 0.093 | 1.098 | 1.000 | 1.205 | 0.166 | -0.024 | 0.976 | 0.886 | 1.076 | -0.117 |
| **Smoking causes lung cancer** | | | | | | | | | | | |

*(Continued)*

**Table 2.** (Continued)

| Explanatory variables | GATS-1 | | | | | GATS-2 | | | | | Change |
| --- | --- | --- | --- | --- | --- | --- | --- | --- | --- | --- | --- |
| | P | β | AOR | 95% C.I. of AOR | | P | β | AOR | 95% C.I. of AOR | | β gats-1 & β gats-2 |
| | | | | Lower | Upper | | | | Lower | Upper | |
| (Yes)[①] | | | 1 | | | | | | 1 | | |
| No | 0.06 | 0.104 | 1.110 | 0.925 | 1.332 | 0.049 | 0.247 | 1.281 | 1.093 | 1.500 | 0.143 |
| **Smokeless tobacco use causes serious illness** | | | | | | | | | | | |
| (Yes)[①] | | | 1 | | | | | | 1 | | |
| No | 0.084 | 0.156 | 1.168 | 0.964 | 1.417 | 0.036 | 0.188 | 1.207 | 1.013 | 1.438 | 0.032 |
| **Intercept** | | -1.184 | | | | | -0.903 | | | | |

Note: P- Proportion of the population; AOR–Adjusted Odds Ratio; C.I.—Confidence interval;

[①] = Reference category.

The population with no knowledge of smoking association with stroke, and lung cancer had higher levels of smoking. In GATS-2 time period, smoking was significantly higher among those who are not aware about smoking causes lung cancer.

## Multivariable binary logistic regression for smokeless tobacco use

Table 3 depicts the results of the multivariable binary logistic regression model to understand the role of different covariates on smokeless tobacco consumption in India at two-survey points of time. The proportion (P), β coefficients and adjusted odds ratio along with a 95 percent CI estimate of smokeless tobacco users according to the various covariates is presented in Table 3. The last column of this table presents the difference in the regression coefficients during the seven-year period.

The coefficients in all the regions were found positive in both the rounds of the survey which indicates that adults from these regions had a higher rate of smokeless tobacco consumption compared to the northern region.

In 2016–17, individuals belonging to all the better off wealth index groups (poorer, middle, richer, richest) had lower levels of smokeless tobacco use as compared to the poorest. The result indicates that women have lower levels of smokeless tobacco use as compared to men in both the rounds of the survey. The rate of smokeless tobacco use in 2009–10 was higher in all age groups as compared to those in the '15–24 years age group' category. As age advances, the levels of consumption also increase. This trend is witnessed in both the rounds of the survey.

Education showed the expected inverse relationship with smokeless tobacco use. The difference in the rate of smokeless tobacco use between uneducated and educated categories becomes more prominent during the inter-survey period. It reflects that the impact of education on smokeless tobacco consumption had become stronger in 2016–17 than in 2009–10.

The table shows that in 2009–10, all occupation groups had lower levels of smokeless tobacco use than the government and non-government employees' group. However, the rate of smokeless tobacco consumption has increased in the self-employed and homemaker groups during the inter-survey period. The differentials between the self-employed and 'government and non-government employees' categories have widened in the year 2016–17 than in 2009–10.

In the first round of the survey, the rate of smokeless tobacco consumption was higher in the population with no knowledge of smokeless tobacco association with serious illnesses than those who were aware. Those who did not have knowledge of smoking association with stroke and heart attack had higher levels of smokeless tobacco use in both the rounds of the survey.

**Table 3. Multivariable binary logistic regression for smokeless tobacco use in India, GATS-2009-10 and GATS-2016-17.**

| Explanatory variables | GATS-1 | | | | | GATS-2 | | | | | Change |
|---|---|---|---|---|---|---|---|---|---|---|---|
| | P | β | AOR | 95% C.I. of AOR | | P | β | AOR | 95% C.I. of AOR | | β gats-1 & β gats-2 |
| | | | | Lower | Upper | | | | Lower | Upper | |
| **Region** | | | | | | | | | | | |
| (North)[①] | | | 1 | | | | | 1 | | | |
| Central | 0.357 | 1.418 | 4.129 | 3.731 | 4.570 | 0.345 | 1.453 | 4.276 | 3.911 | 4.676 | 0.035 |
| East | 0.305 | 1.724 | 5.607 | 5.069 | 6.201 | 0.279 | 1.515 | 4.548 | 4.153 | 4.980 | -0.209 |
| Northeast | 0.044 | 1.877 | 6.533 | 5.958 | 7.163 | 0.067 | 2.097 | 8.143 | 7.472 | 8.875 | 0.220 |
| West | 0.162 | 1.034 | 2.813 | 2.532 | 3.124 | 0.174 | 1.105 | 3.019 | 2.736 | 3.330 | 0.071 |
| South | 0.119 | 0.248 | 1.281 | 1.147 | 1.430 | 0.109 | 0.127 | 1.136 | 1.027 | 1.256 | -0.120 |
| **Type of Residence** | | | | | | | | | | | |
| (Urban)[①] | | | 1 | | | | | 1 | | | |
| Rural | 0.8 | 0.057 | 1.059 | 1.000 | 1.121 | 0.751 | 0.029 | 1.030 | 0.975 | 1.088 | -0.028 |
| **Wealth index quintiles** | | | | | | | | | | | |
| (Poorest)[①] | | | 1 | | | | | 1 | | | |
| Poorer | 0.304 | 0.046 | 1.047 | 0.960 | 1.141 | 0.292 | -0.091 | 0.913 | 0.857 | 0.973 | -0.137 |
| Middle | 0.203 | -0.106 | 0.900 | 0.822 | 0.984 | 0.186 | -0.175 | 0.839 | 0.781 | 0.902 | -0.069 |
| Fourth | 0.149 | -0.404 | 0.667 | 0.608 | 0.733 | 0.17 | -0.380 | 0.684 | 0.633 | 0.739 | 0.025 |
| Richest | 0.09 | -0.763 | 0.466 | 0.421 | 0.517 | 0.081 | -0.814 | 0.443 | 0.404 | 0.486 | -0.052 |
| **Age groups** | | | | | | | | | | | |
| (15–24)[①] | | | 1 | | | | | 1 | | | |
| 25–44 | 0.458 | 0.440 | 1.553 | 1.432 | 1.684 | 0.483 | 0.562 | 1.755 | 1.612 | 1.910 | 0.122 |
| 45–64 | 0.264 | 0.423 | 1.527 | 1.394 | 1.673 | 0.257 | 0.627 | 1.871 | 1.709 | 2.050 | 0.203 |
| 65 and above | 0.09 | 0.452 | 1.571 | 1.384 | 1.783 | 0.092 | 0.608 | 1.836 | 1.654 | 2.038 | 0.156 |
| **Gender** | | | | | | | | | | | |
| (Male)[①] | | | 1 | | | | | 1 | | | |
| Female | 0.407 | -0.571 | 0.565 | 0.527 | 0.605 | 0.334 | -0.725 | 0.484 | 0.456 | 0.514 | -0.154 |
| **Education** | | | | | | | | | | | |
| (No formal schooling)[①] | | | 1 | | | | | 1 | | | |
| Upto Primary | 0.275 | -0.010 | 0.990 | 0.923 | 1.061 | 0.267 | -0.045 | 0.956 | 0.898 | 1.018 | -0.034 |
| Upto Secondary | 0.223 | -0.199 | 0.819 | 0.760 | 0.883 | 0.274 | -0.280 | 0.756 | 0.708 | 0.808 | -0.080 |
| Above Secondary | 0.085 | -0.551 | 0.576 | 0.525 | 0.633 | 0.096 | -0.754 | 0.470 | 0.431 | 0.513 | -0.203 |
| **Occupation** | | | | | | | | | | | |
| (Government and non-government employee)[①] | | | 1 | | | | | 1 | | | |
| Self employed | 0.373 | -0.112 | 0.894 | 0.839 | 0.953 | 0.605 | 0.143 | 1.154 | 1.072 | 1.243 | 0.255 |
| Student | 0.03 | -1.057 | 0.347 | 0.302 | 0.400 | 0.018 | -1.147 | 0.318 | 0.272 | 0.371 | -0.090 |
| Homemaker | 0.244 | -0.577 | 0.562 | 0.516 | 0.611 | 0.19 | -0.480 | 0.619 | 0.564 | 0.678 | 0.097 |
| Retired or unemployed | 0.068 | -0.130 | 0.878 | 0.784 | 0.983 | 0.078 | -0.155 | 0.856 | 0.769 | 0.953 | -0.025 |
| **Smoking causes serious illness** | | | | | | | | | | | |
| (Yes)[①] | | | 1 | | | | | 1 | | | |
| No | 0.124 | -0.414 | 0.661 | 0.490 | 0.891 | 0.077 | -0.162 | 0.850 | 0.774 | 0.934 | 0.252 |
| **Smoking causes stroke** | | | | | | | | | | | |
| (Yes)[①] | | | 1 | | | | | 1 | | | |
| No | 0.273 | 0.137 | 1.147 | 1.072 | 1.228 | 0.284 | 0.239 | 1.270 | 1.190 | 1.356 | 0.102 |
| **Smoking causes heart attack** | | | | | | | | | | | |
| (Yes)[①] | | | 1 | | | | | 1 | | | |
| No | 0.186 | 0.169 | 1.184 | 1.094 | 1.282 | 0.202 | 0.134 | 1.143 | 1.058 | 1.236 | -0.035 |
| **Smoking causes lung cancer** | | | | | | | | | | | |

(Continued)

**Table 3.** (Continued)

| Explanatory variables | GATS-1 | | | | | GATS-2 | | | | | Change |
|---|---|---|---|---|---|---|---|---|---|---|---|
| | P | β | AOR | 95% C.I. of AOR | | P | β | AOR | 95% C.I. of AOR | | β gats-1 & β gats-2 |
| | | | | Lower | Upper | | | | Lower | Upper | |
| (Yes)[①] | | | 1 | | | | | 1 | | | |
| No | 0.034 | -0.300 | 0.741 | 0.634 | 0.867 | 0.057 | 0.071 | 1.073 | 0.948 | 1.215 | 0.370 |
| **Smokeless tobacco use causes serious illness** | | | | | | | | | | | |
| (Yes)[①] | | | 1 | | | | | 1 | | | |
| No | 0.059 | 0.194 | 1.215 | 1.031 | 1.430 | 0.043 | 0.060 | 1.061 | 0.920 | 1.224 | -0.135 |
| **Intercept** | | **-1.861** | | | | | **-2.150** | | | | |

Note: P—Proportion of the population; AOR–Adjusted Odds Ratio; C.I.—Confidence interval;

[①] = Reference category.

In 2009–10, the population who did not have knowledge of smoking association with lung cancer had significantly lower levels of smokeless tobacco use. However, in the same time period the levels of consumption have increased among the people were aware about smokeless tobacco use causes serious illness, indicating the importance of health-related knowledge on tobacco consumption.

## Decomposition of change in smoking by using multivariable binary logistic regression model in India, 2009–10 and 2016–17

Table 4 depicts the decomposition of the overall decline in the smoking form of tobacco consumption into different components, namely—rate, composition and interaction. The result in this Table is based on the coefficients of multivariable binary logistic regression and proportional distribution of population reported in Table 2. It is evident from the Table that the leading components of decline in the level of smoking is propensity and interaction, which explains around 41 per cent and 42 per cent of the overall smoking consumption change respectively. Interaction is an inter-play of the rate and composition components—(interaction at aggregate and sub-group levels). Around 17 per cent of the overall decline is being explained by a shift in the population-composition component.

Further, this table indicates that place of residence and occupation of the respondent covariates contribute significantly in reducing the prevalence of smoking during the seven-year period, regardless of change in the composition of population. The positive sign of the propensity factors found in the above mentioned covariates indicates that the rate of smoking form of tobacco use has declined more among their subgroups than their reference category. Occupation and type of residence added around 104 and 35 per cent respectively to the overall change in smoking, keeping the composition of population as a constant. The contribution of the self-employed and homemakers in reducing smoking was relatively high from government and non-government employees.

The negative sign in the sub-group who did not have knowledge of smoking causing heart attack leads to the proposition that this group did not contribute to the decline in smoking with respect to the reference category. This implies that those having knowledge that smoking is associated with heart attack contributed more to lowering the rate of smoking. Those who were unaware that smoking causes stroke and lung cancer contributed around two and six per cent, respectively while those unaware that smokeless tobacco causes serious illness contributed around two percent to the total decline in smoking.

**Table 4. Decomposition of change in smoking by using multivariable binary logistic regression model in India, 2009–10 and 2016–17.**

| Explanatory variables | Percentage change due to | | |
|---|---|---|---|
| | Rate | Composition | Interaction |
| **Intercept** | **501.96** | | |
| **Region** | | | |
| (North)[①] | | | |
| Central | -136.40 | 4.67 | 24.58 |
| East | -95.00 | -0.69 | -4.02 |
| Northeast | -2.08 | 2.27 | -0.31 |
| West | -20.25 | 12.37 | 3.87 |
| South | -50.61 | 2.92 | 2.18 |
| **Total** | **-304.34** | **21.54** | **26.29** |
| **Type of Residence** | | | |
| (Urban)[①] | | | |
| Rural | 34.75 | -0.33 | -1.22 |
| **Total** | **34.75** | **-0.33** | **-1.22** |
| **Wealth index quintiles** | | | |
| (Poorest)[①] | | | |
| Poorer | -49.48 | -0.28 | 0.39 |
| Middle | -38.18 | -1.35 | 2.78 |
| Fourth | -42.24 | 0.35 | -1.67 |
| Richest | -40.69 | 0.56 | 3.33 |
| **Total** | **-170.59** | **-0.72** | **4.82** |
| **Age groups** | | | |
| (15–24)[①] | | | |
| 25–44 | -21.30 | 34.78 | -3.45 |
| 45–64 | -9.98 | -62.58 | 2.23 |
| 65 and above | -10.38 | 1.83 | -0.28 |
| **Total** | **-41.65** | **-25.97** | **-1.50** |
| **Gender** | | | |
| (Male)[①] | | | |
| Female | -9.40 | 10.41 | 0.54 |
| **Total** | **-9.40** | **10.41** | **0.54** |
| **Education** | | | |
| (No formal schooling)[①] | | | |
| Upto Primary | -15.07 | 3.02 | 0.93 |
| Upto Secondary | -15.57 | -16.12 | -3.08 |
| Above Secondary | -20.78 | -3.62 | -1.19 |
| **Total** | **-51.42** | **-16.72** | **-3.34** |
| **Occupation** | | | |
| (Government and non-government employee)[①] | | | |
| Self employed | 80.80 | -5.49 | 42.54 |
| Student | 0.08 | -0.57 | 0.01 |
| Homemaker | 17.84 | 4.56 | -4.27 |
| Retired or unemployed | 5.33 | -0.97 | 0.99 |
| **Total** | **104.05** | **-2.48** | **39.27** |
| **Smoking causes serious illness** | | | |
| (Yes)[①] | | | |
| No | 4.16 | 5.82 | -1.91 |

*(Continued)*

**Table 4.** (Continued)

| Explanatory variables | Percentage change due to | | |
|---|---|---|---|
| | *Rate* | *Composition* | *Interaction* |
| **Total** | **4.16** | **5.82** | **-1.91** |
| **Smoking causes stroke** | | | |
| (Yes)[①] | | | |
| No | 2.37 | -8.49 | -0.57 |
| **Total** | **2.37** | **-8.49** | **-0.57** |
| **Smoking causes heart attack** | | | |
| (Yes)[①] | | | |
| No | -15.59 | -2.21 | 2.78 |
| **Total** | **-15.59** | **-2.21** | **2.78** |
| **Smoking causes lung cancer** | | | |
| (Yes)[①] | | | |
| No | 5.66 | -0.75 | -1.04 |
| **Total** | **5.66** | **-0.75** | **-1.04** |
| **Smokeless tobacco use causes serious illness** | | | |
| (Yes)[①] | | | |
| No | 1.77 | -4.94 | -1.01 |
| **Total** | **1.77** | **-4.94** | **-1.01** |
| **Grand total** | **41.24** | **16.60** | **42.17** |

[①] = Reference category.

The signs of negativity propensity were found in the case of covariates of region, wealth index, age groups, gender, and education. It can be inferred that those living in northern India, poorest population, 15–24 age group, males, and government and non-government employees were having lower rate of smoking in the year 2016–2017 in comparison to the earlier period 2009–2010.

The decline in the level of smoking that took place due to shifts in population structure is explained by region and gender keeping the rate of GATS-1 as a constant. Regional changes explained around 22 per cent of the total change in smoking regardless of the change in smoking behaviour within the regions. Most of the change has occurred because of a shift of population from the northern region towards the western and central regions. Similarly, the change that has occurred in the shift of composition of gender (females) also favours the decline in smoking.

Most of the changes in the interaction of rate and population structure that have occurred during the seven-year period have favoured a decline in the level of smoking. Regions, wealth status, gender, occupation, explain the decline in the level of smoking due to interaction. Occupation explained around 39 per cent of the total change in smoking. The interaction of rate and population composition of the self-employed group resulted in a significant reduction in the level of smoking during the seven years period. Decline in the percentage of smokers in the age group of 45–64 years helped in reducing the level of smoking.

The second foremost component is the interaction that occurred in the regions, which explained around 27 per cent of the overall change in the level of smoking of adults aged 15 years and above during GATS-1 and GATS-2. The interaction of rate and population composition in the central region favours the decline in smoking.

### Decomposition of change in smokeless tobacco consumption using multivariable binary logistic regression model in India, 2009–10 and 2016–17

Table 5 depicts the decomposition of the overall decline in the smokeless form of tobacco consumption into different components, namely—rate, composition and interaction at aggregate and sub-group levels. It is evident from the Table that the leading component of decline in the level of smokeless tobacco is change in propensity, which explains around 81 per cent, eight per cent of the overall decline is being explained by a shift in the population-composition component and the interaction contributed by around 11 per cent.

Further, this table indicates that the covariates of age group and occupation of the respondent contribute significantly in reducing the prevalence of smokeless tobacco consumption during the seven-year period, regardless of change in the composition of population. Age group and occupation added around 18 and 17 per cent respectively to the overall change in smokeless tobacco consumption, keeping the composition of population as a constant.

Further, the negative sign in the sub-group who did not have knowledge of smoking association with heart attack and smokeless tobacco association with serious illnesses' leads to the proposition that this group did not contribute to the decline in smokeless tobacco use with respect to the reference category. This implies that those having knowledge that smokeless tobacco causes serious illnesses contributed more to lowering the rate of smokeless tobacco use. Those who were unaware that smoking causes serious illnesses, stroke, and lung cancer contributed around five, four and two per cent respectively to the decline in smokeless tobacco use.

Most of the decline in the level of smokeless tobacco use that took place due to shifts in population structure is explained by gender, occupation, and age keeping the rate of GATS-1 as a constant. The foremost component of composition is the shift in the structure of the gender of the population (towards female), which explained around six per cent of the overall change in the level of smokeless tobacco consumption of adults aged 15 years and above during GATS-1 and GATS-2. Most of the change has occurred because of an increase in population from the male towards female category. Similarly, the change that has occurred in the shift of composition of occupation of the population (towards homemakers) favours the decline in smokeless tobacco consumption.

## Discussion

Smoking and smokeless tobacco consumption imposes extensive burden of disease and death on the public health. The Government of India has undertaken various initiatives and legislation aimed at tobacco control. The Cigarettes and Other Tobacco Products (Prohibition of Advertisement and Regulation of Trade and Commerce, Production, Supply and Distribution) Act (COTPA) came into force in 2003 [18], making it the principal comprehensive law governing tobacco control in India. Some of the rules promulgated under this law were prohibition of direct and indirect advertisements of tobacco products, sale of tobacco to minors, smoking in public places, and within a radius of 100 yards of educational institutions [19]. It also included mandatory display of pictorial warning on tobacco product packages, testing of tar and nicotine content of all tobacco products. These rules faced numerous socio-political and legal blockades, following which the Revised Smoke-free Rules came into effect from 2008 [20]. The Government of India ratified the WHO Framework Convention on Tobacco Control (WHO FCTC) in 2004, which enlists key strategies for reduction in demand and supply of tobacco [21]. Further, to strengthen implementation of the tobacco control provisions under COTPA and the WHO FCTC, the Government of India piloted the National Tobacco Control

**Table 5. Decomposition of change in smokeless tobacco consumption using multivariable binary logistic regression model in India, 2009–10 and 2016–17.**

| Explanatory variables | Percentage change due to | | |
|---|---|---|---|
| | *Rate* | *Composition* | *Interaction* |
| **Intercept** | **71.14** | | |
| **Region** | | | |
| (North)[①] | | | |
| Central | 1.84 | -2.50 | -0.06 |
| East | -9.37 | -6.59 | 0.80 |
| Northeast | 1.42 | 6.35 | 0.74 |
| West | 1.69 | 1.82 | 0.13 |
| South | -2.12 | -0.36 | 0.18 |
| **Total** | **-6.54** | **-1.28** | **1.78** |
| **Type of Residence** | | | |
| (Urban)[①] | | | |
| Rural | -3.29 | -0.41 | 0.20 |
| **Total** | **-3.29** | **-0.41** | **0.20** |
| **Wealth index quintiles** | | | |
| (Poorest)[①] | | | |
| Poorer | -6.12 | -0.08 | 0.24 |
| Middle | -2.06 | 0.26 | 0.17 |
| Fourth | 0.53 | -1.25 | 0.07 |
| Richest | -0.67 | 1.01 | 0.07 |
| **Total** | **-8.33** | **-0.05** | **0.56** |
| **Age groups** | | | |
| (15–24)[①] | | | |
| 25–44 | 8.21 | 1.62 | 0.45 |
| 45–64 | 7.92 | -0.44 | -0.21 |
| 65 and above | 2.06 | 0.13 | 0.05 |
| **Total** | **18.19** | **1.31** | **0.28** |
| **Gender** | | | |
| (Male)[①] | | | |
| Female | -8.91 | 6.13 | 1.60 |
| **Total** | **-8.91** | **6.13** | **1.60** |
| **Education** | | | |
| (No formal schooling)[①] | | | |
| Upto Primary | -1.21 | 0.01 | 0.04 |
| Upto Secondary | -2.66 | -1.49 | -0.61 |
| Above Secondary | -2.49 | -0.89 | -0.32 |
| **Total** | **-6.35** | **-2.37** | **-0.89** |
| **Occupation** | | | |
| (Government and non-government employee)[①] | | | |
| Self employed | 13.82 | -3.82 | 8.59 |
| Student | -0.41 | 1.86 | 0.16 |
| Homemaker | 3.48 | 4.58 | -0.77 |
| Retired or unemployed | -0.30 | -0.19 | -0.04 |
| **Total** | **16.58** | **2.43** | **7.94** |
| **Smoking causes serious illness** | | | |
| (Yes)[①] | | | |
| No | 4.63 | 2.86 | -1.75 |

(*Continued*)

**Table 5.** (*Continued*)

| Explanatory variables | Percentage change due to | | |
| --- | --- | --- | --- |
| | *Rate* | *Composition* | *Interaction* |
| **Total** | **4.63** | **2.86** | **-1.75** |
| **Smoking causes stroke** | | | |
| (Yes)[①] | | | |
| No | 4.13 | 0.22 | 0.17 |
| **Total** | **4.13** | **0.22** | **0.17** |
| **Smoking causes heart attack** | | | |
| (Yes)[①] | | | |
| No | -1.07 | 0.40 | -0.09 |
| **Total** | **-1.07** | **0.40** | **-0.09** |
| **Smoking causes lung cancer** | | | |
| (Yes)[①] | | | |
| No | 1.85 | -1.01 | 1.25 |
| **Total** | **1.85** | **-1.01** | **1.25** |
| **Smokeless tobacco use causes serious illness** | | | |
| (Yes)[①] | | | |
| No | -1.16 | -0.46 | 0.32 |
| **Total** | **-1.16** | **-0.46** | **0.32** |
| **Grand total** | **80.88** | **7.76** | **11.36** |

[①] = Reference category.

Programme (NTCP) in 2007–2008 [8]. From their inception, these tobacco control initiatives have evolved and expanded across the country. The inter survey period of GATS witnessed the emergence of additional interventions from both, the central and state governments resulting in powerful mechanisms for tobacco control. These include Food Safety and Standards Authority of India (FSSAI) prohibition regulations for *gutka*, steep excise duties on tobacco products, judicial clarifications on regulations pertaining to tobacco product bans and mandatory 85% graphic health warnings on all tobacco product packages [22, 23].

Result shows that the level of smoking and smokeless tobacco consumption has declined in India from 2009–10 to 2016–17, but there are differentials in consumption. These variations are generally viewed in terms of socio-economic variables such as education, gender, age, economic condition and place of residence. Previous studies have shown that the tobacco consumption is disproportionately higher among lower socio-economic groups, manifested in the lower age of initiation [11], and lower quit rates among these groups [24]. In addition, in India, culture plays an important role in influencing the type and pattern of tobacco use. The decline in tobacco use could be the result from a decrease in use among a particular socio-economic group or a change in the population composition of the same group. Therefore, with the help of nationally representative GATS-1 and GATS-2 database, this study tries to examine the contribution of such factors to the change in smoking and smokeless tobacco use in India.

The central finding of the study is that the propensity component is primarily responsible for major tobacco consumption decline. For smoking, and smokeless tobacco the propensity component explained about 41 per cent and 81 per cent respectively. Moreover, the composition component contributed about 17 per cent for smokers and eight per cent for smokeless tobacco use, of the total change. The inter-play of propensity and population composition contributed 42 and 11 percent for change in smoking and smokeless tobacco use respectively.

Previous studies have documented regional variations in all forms of tobacco use, which may be due to contextual factors such as the social environment (deprivation, area-level mean income, area-level income inequality and social capital), shared cultural and social norms regarding tobacco use and the availability and implementation of tobacco control policies in a given area [25]. The results of this paper based on GATS-2009-10 show that compared to the northern region, all the other regions had lower levels of smoking (excluding northeast). This trend has continued during the seven-year period, with the northern region showing higher smoking in 2016–17. Similar trends of smoking being higher in the northern and northeast regions in the country has been documented in past studies [10, 12]. One of the factors contributing to this trend in the above mentioned regions was the higher prevalence of smoking among women [8, 12, 26, 27]. Apart from tradition, politico-legal and geographical factors, the extent to which each state has been able to implement the anti-tobacco measures plays a pivotal role in inter-state and regional variations [8, 28]. This persisting regional trend highlights the need for more intensive and gender sensitive tobacco cessation interventions in north and northeast India.

Health being a subject in the State List in India [29], has led to the development of state legislations and programmes for tobacco control based on the socio-political, economic and cultural context. These regional variations could be explained due to the above mentioned inter-state differences in successful implementation of tobacco control initiatives. For instance, Rajasthan, a state in the central region levies the highest tax on all tobacco products [19]. Additionally, it launched several innovative campaigns against tobacco consumption at places like schools, colleges, police stations and government offices, culminating in the state being the recipient of the WHO Tobacco Control Award in 2019 [30]. According to the WHO Global Tobacco Epidemic Report 2017, big cities such as Kanpur, Lucknow from the state of Uttar Pradesh and Jaipur from Rajasthan (central region), and Kolkata from West Bengal (eastern region) have achieved high levels of coverage in tobacco control measures such as awareness of dangers of tobacco use and availability of help to quit [31]. Moreover, in 2013, Bihar, a state in eastern India carried out an excellent example of tobacco cessation intervention outside the health sector comprising of educational efforts, tobacco control policies and cessation support [32, 33]. These rigorous and innovative campaigns by different states in the eastern and central region can explain the higher rates of decline in these regions in smoking and smokeless tobacco consumption.

The contribution of the 25–44 years age group to decline in smokeless tobacco use related to rate and composition is around eight and two percent respectively. In both the rounds of GATS, the age groups of 25 years and above had higher levels of smokeless tobacco use. The contribution of age in reducing smokeless tobacco use declines with increasing age. Previous studies have also shown the same trend of an increase in smokeless tobacco usage with age [34]. This could be due to people adopting the habit of smokeless tobacco use by the time a particular age is reached and lower quitting rate due to its addictive nature. Therefore, it is important that tobacco control policies target the adolescents and younger age groups that can improve the knowledge and awareness of the ill effects of tobacco in this group. Strategies to prevent the adoption of smoking and smokeless tobacco use in the first place are more effective and successful than tobacco cessation measures in reducing the overall prevalence of tobacco consumption [35].

The wealth status of household is a significant predictor of smoking. Our results based on disaggregated data on smoking and smokeless tobacco use shows that the smoking levels declined with improving wealth status indicated by lower levels of smoking in the middle, richer and richest wealth population groups in 2016–17. A study conducted by Singh *et al.* (2015) found that cigarette smoking was positively associated with household wealth and the

richest category had higher odds of smoking cigarettes [3.86 (95% CI: 2.54–5.86)] relative to the poorest group. It can be inferred that the better off population (richer and richest) realised the harmful effects of smoking leading to a decline in their consumption. However, the poorer population groups continue to smoke *bidis* [36]. India has historically had absent or lower taxes on *bidis* and a complex system of taxing cigarettes resulting in lower prices. Another possible factor that can explain the higher smoking in the poorer groups is the differential treatment of the taxation policy towards *bidis*. For instance, in the fiscal year 2015–16, the central excise tax was 16 rupees (US$0.22) per 1000 handmade *bidi* sticks while it was 3790 rupees (US$51.4) for 1000 cigarettes of lengths of 75 mm and over [37]. This has resulted in the price of *bidis* being as low as 0.20 rupees (US$0.0027), and pack prices between three (US$0.041) and 20 rupees (US$0.27) across India [38]. Further, a rise in disposable incomes in India could have also lead to an increase in cigarette consumption among the poorer groups [39]. Cigarettes become affordable when they are sold as loose sticks and not in a pack. The poorer population can therefore, in addition to *bidis*, purchase and consume loose cigarette sticks resulting in modest declines in their consumption during the inter-survey period. Studies conducted across countries have shown that cigarette consumers in low-and middle-income countries are more sensitive (reduction in smoking) to an increase in the price as compared to the high-income countries [40]. Higher taxation of tobacco products has been established as the single most effective intervention to reduce consumption [41]. Despite efforts of the government to increase the taxes on tobacco products, India is still lower than the WHO's limit of 75 percent on retail price. This affordability and differential pricing has lead to widespread use of tobacco products. For smokeless tobacco consumption as well, wealth status is a significant predictor during GATS-1 and GATS-2. Studies have proven an inverse association between household wealth status and smokeless tobacco use [42]. This study shows that in both the rounds of GATS, almost all better-off wealth quintile groups had lower levels of smokeless tobacco use as compared to the poorest.

The contribution of the self-employed and homemaker groups to the decline in smoking related to rate is around 81 and 18 per cent respectively. For decline in smokeless tobacco use, the self-employed and homemakers contributed 14 and three per cent respectively.

Women contributed less to the decline in smoking as compared to men. Earlier studies have shown that minority proportion of women access tobacco cessation clinics and out of this, most of these women are smokeless tobacco users [43]. It is known that women smoking tobacco is culturally unacceptable in India [44], thereby creating an obstacle to avail help for quitting. Hence, it is essential to make the NTCP more gender inclusive so that women can avail tobacco cessation services easily. Women had lower levels of smokeless tobacco consumption in GATS-1 and GATS-2 and contributed lesser to the decline in use as compared to men. This decline can be attributed to policy measures such as a blanket ban imposed by the government on production, storage, or distribution of all forms of chewing tobacco products including zarda and pan masala in 2011 [45].

The population with knowledge of smokeless tobacco association with serious illnesses contributed more to the decline in smokeless tobacco use as compared to those who did not know. Those who did not have knowledge of the association of smoking with serious illness and specific conditions (stroke, lung cancer) contributed more to the decline in smoking that those who had the knowledge. This knowledge of tobacco causing specific diseases and ailments has not translated into encouraging practices that lead to decline in the levels of smoking and smokeless tobacco consumption. This indicates potential for improvement in the mass media campaigns especially those highlighting the specific diseases and ill effects of smoking and smokeless tobacco.

Learning from experiences of other low-and-middle income countries such as Nepal, to enforce more effective tobacco control it is important for developing countries like India to engage with politicians, legislators, the media, civil society and raise awareness among citizens regarding the tobacco lobby as well as different provisions of tobacco control [46].

This paper has a few limitations. First, the self-reported nature of the data collected in GATS may lead to underestimation of tobacco prevalence. Second, certain predictor variables included in the second round of GATS were not included in the first, which prevented a comparative analysis and subsequently led to exclusion of those variables from the analysis. Third, the study does not document the decline in smoking and smokeless tobacco use separately categorised for the array of products consumed under the above two forms due to a small sample size. Fourth, though we have tried to identify different programmes and interventions and their impact on tobacco use. However, direct linkage could not be established since the data collected in GATS does not reflect the effect on actual quitting. Moreover, to understand the reasons underlying the identified pattern of tobacco consumption and the barriers and facilitators for cessation of tobacco, contextually relevant qualitative research studies must be designed.

## Conclusion

India faces a high burden of tobacco consumption. Different factors influence the prevalence of smoking and smokeless tobacco use. In addition to socio-economic inequalities, regional inequalities must be monitored. The northern and north-eastern region need more focus on tobacco control programmes since they show a trend of higher tobacco burden over time. Moreover, the population that is aware of the ill effects of tobacco on health mostly have a relatively lower contribution as compared to the unaware, to the declining tobacco consumption. Since 2016, the government has allowed warnings to cover 85% of the front and back of the package, and a combination of graphic and regional language text warnings are currently used in the country. Studies have documented that these written statutory warnings are predominantly in English and Hindi [47–49]. Keeping in mind the linguistic diversity influenced by regional variations and socio-economic inequalities in tobacco use in India, all campaigns and warnings on product packages must be in the local language or dialect enabling the user to clearly comprehend the damaging effect of consuming tobacco.

Smokeless tobacco use in work and public places must be prohibited akin to the smoking ban. To achieve the Indian health target that aims to reduce tobacco use by 15% by 2020 and by 30% by 2025 [6], culture and context specific strategies addressing the inequalities in tobacco use must be devised, accompanied by strict implementation of the tobacco control policies.

## Supporting information

**S1 File.**
(DOCX)

## Author Contributions

**Conceptualization:** Priyanka Dixit.

**Data curation:** Supriya Lahoti.

**Formal analysis:** Supriya Lahoti.

**Methodology:** Priyanka Dixit.

**Software:** Supriya Lahoti.

**Supervision:** Priyanka Dixit.

**Validation:** Priyanka Dixit.

**Visualization:** Supriya Lahoti.

**Writing – original draft:** Supriya Lahoti.

**Writing – review & editing:** Priyanka Dixit.

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
