## [Decision Letter · Decision Letter 0]

28 Jul 2020

PONE-D-20-16653

Declining Trend of Smoking and Smokeless Tobacco in India: A Decomposition Analysis

PLOS ONE

Dear Dr. DIXIT,

Thank you for submitting your manuscript to PLOS ONE. After careful consideration, we feel that it has merit but does not fully meet PLOS ONE’s publication criteria as it currently stands. Therefore, we invite you to submit a revised version of the manuscript that addresses the points raised during the review process.

We look forward to receiving your revised manuscript.

Kind regards,

Stanton A. Glantz

Academic Editor

PLOS ONE

Journal Requirements:

2.PLOS requires an ORCID iD for the corresponding author in Editorial Manager on papers submitted after December 6th, 2016. Please ensure that you have an ORCID iD and that it is validated in Editorial Manager. To do this, go to ‘Update my Information’ (in the upper left-hand corner of the main menu), and click on the Fetch/Validate link next to the ORCID field. This will take you to the ORCID site and allow you to create a new iD or authenticate a pre-existing iD in Editorial Manager. Please see the following video for instructions on linking an ORCID iD to your Editorial Manager account: https://www.youtube.com/watch?v=_xcclfuvtxQ

<h1>** **</h1>

Reviewers' comments:

Reviewer's Responses to Questions

**Comments to the Author**

1. Is the manuscript technically sound, and do the data support the conclusions?

Reviewer #1: Partly

Reviewer #2: Yes

Reviewer #3: Partly

2. Has the statistical analysis been performed appropriately and rigorously? 

Reviewer #1: No

Reviewer #2: I Don't Know

Reviewer #3: No

3. Have the authors made all data underlying the findings in their manuscript fully available?

Reviewer #1: Yes

Reviewer #2: No

Reviewer #3: Yes

4. Is the manuscript presented in an intelligible fashion and written in standard English?

Reviewer #1: Yes

Reviewer #2: Yes

Reviewer #3: Yes

5. Review Comments to the Author

Reviewer #1: Introduction section is lengthy, please concise it.

Authors only focused on community or individual level determinant factor of smoking but tobacco industry activities, tobacco control policies and its implementation are also crucial on tobacco control.

Authors provided several interventions during the study period from 2009 to 2016 and the prevalence of tobacco use declined from 35% to 29%. If authors can provide previous than this study period declining trend on tobacco use in India, it is good to know the effects of these interventions. I am unable to read reference no 15.

Last two paragraph of Introduction section needs to revise. Authors can provide details of the analysis in supplemental file. Rationale is not much convincing why this study had done, why decomposition analysis, what are the difference compared to other analysis. If there any study used other analysis approach and had limitations or any discrepancy.

Who supported GATS in India, who was the implementation agency?

What does it mean by Indian residents? Citizen or any who reside in India for certain time.

Sampling details is not available, how the household was selected? How individual was selected in the household?

What was the response rate for both period?

Please delete from Line 108 to 110. Just provide the references in the text and authors need to provide details in the study.

Line 112 to 115 is useless. In fact, the variables were not selected on the basis of the literature, and the survey only included demographic variables.

Line 116 to 125 is not useful.

I don’t think caste and religion are household level category. Religion and caste should be an individual choice. What is backward classes and scheduled caste?

How many household assets were included for wealth index?

Line 140, what are the common variables? Please provide details about the variables only which you had used in your study.

It would be good to see any tobacco use in the dependent variable.

What is the reference category for smoking only and smokeless only? It included never user or former and never user.

How they calculated percent relative change? Was it a weighted percent?

Line 155 to 173 is very hard to understand. It is better to specify different analytic approaches separately. For example, why, when and how authors used logistic (may be) regression model? What they did with this regression model? So on….. I don’t think authors did multivariate analysis.

It is better to provide details about decomposition analysis in supplemental file. Authors can provide step by step process of this analysis with SPSS syntax.

Please specify, how the survey weights were used in the study?

It is not clear whether authors combined both survey or analyzed separately, if combined how they treated survey weights.

How the interaction worked in bivariate regression model? How propensity matched in bivariate model? Which propensity match was used?

What was the accuracy of the decomposition model?

Explain your model with Rate, Composition and Interaction.

Please specify proportion of the population in Tables 2 and 3. How it was calculated?

Line 201, …..and dual use tobacco…… I didn’t see the dual use tobacco in tables and Methods, how dual use was defined? and why not presented in the results?

Very hard to follow Methods and Results section. Tables title shows multivariate analysis but Methods section did not mention about it.

Interpretation of results is very hard to understand, please make it simple and clear. Authors can focus on major significant results on their interpretation.

It would be better to put tables 2 and 3 in supplemental files. In addition to tables 4 and 5, I would suggest another table for Any Tobacco use.

How Grand total was calculated in Tables 4 and 5. If the tables 4 and 5 outcomes were from bivariate analysis, how could you eliminate the variable selection bias?

First paragraph of the Discussion section is repeated literature from the Introduction.

Dual tobacco uses again came in the Discussion but there is no results about dual use tobacco.

Please provide the references for Line 461 to 464, please provide these regions means which one…what interventions are available? Author should discuss potential contributing factors for higher and lower regional reduction in tobacco use.

Discuss one factor at single place. For example, age is discussed in several paragraphs in the Discussion.

Conclusion is similar to discussion. Please rewrite your conclusion with your study’s main finding.

Please try to include original articles in the references instead internet websites. Some websites are not trusty.

Reviewer #2: This is a well written manuscript on comparing tobacco use prevalence across two waves of Global Adult Tobacco Survey in India. Authors have compared tobacco use prevalence data from GATS-1 and GATS-2 conduced in India to examine the demographic and socioeconomic correlates of

smoking and smokeless tobacco use for both the rounds of the survey.

Please see my feedback below.

General Comments

1. The manuscript may benefit from a good language and grammar editing.

Introduction:

2. line 47-48: ADD Khaini under Smokeless tobacco (ST) use, which is the most prevalent form of ST use in India.

3. Line 75-78: Please refer to these published papers on the theme: Suliankatchi Abdulkader R, Sinha DN, Jeyashree K, et al. Trends in tobacco consumption in India 1987-2016: impact of the World Health Organization Framework Convention on Tobacco Control. Int J Public Health. 2019;64(6):841-851. doi:10.1007/s00038-019-01252-x

Singh A, Arora M, Bentley R, et al Geographic variation in tobacco use in India: a population-based multilevel cross-sectional study BMJ Open 2020;10:e033178. doi: 10.1136/bmjopen-2019-033178

Material and Methods:

4. Role of Geographical areas need to be considered in this analysis, given published literature in india using GATS has highlighted that people’s use of tobacco products varies by local areas (city ward and village) across India and the variation in this clustering by tobacco products.

5. Zones and Region has been used inter-changeably between text and Table1, thus confuses the reader.

Discussion: Overall discussion needs to be revised to make it aligned with results presented in the manuscript as it discusses and makes recommendations on many issues not related to the results of this manuscript.

6. line 548-549: Pack warning in India as per rules and notification are supposed to be in regional languages too. The rules mention that warning language will be as per language used for branding. Please correct this.

7. lines 554-555: Smokeless tobacco is also cheap as it is sold in small sachets. how does affordability of single sale of cigarette compare to sale of bidis and smokeless tobacco? This explanation is not clear and not aligned with results.

8. Lines 562-566: Unsure which result is being discussed here.

Overall check policy development years and details mentioned in Introduction and discussion section of the manuscript.

Reviewer #3: The manuscript entitled ‘Declining Trend of Smoking and Smokeless Tobacco in India: A Decomposition Analysis’ with the aim to examine the socioeconomic correlates and delineate the factors contributing to a change in smoking and smokeless tobacco.

The manuscript requires further improvement.

Comments

Statistical analysis

The description of the statistical analysis requires improvement and reorganization. A subtitle Statistical Analysis to be provided Information on the variable coding to be provided. The word adjusted to be used where applicable. The information on multicollinearity (if any), variable selection method in the analysis, interaction (if any), goodness of fit/model fit etc to be stated.

Line 150 -153, the coding for ‘0’ categories to be provided.

Line 196-197, more information to be provided.

Line 197, typo ‘Statisticsversion’.

Results

Line 213, the results for dual use to be provided in the table(s).

Table 1, SCT, ST and OBC to be denoted in the table footnote. n to be provided apart from %.

Line 235, the sentence ‘The Central region followed by the Western region has attained a higher rate of smoking decline as opposed to other regions’ not clear based on the table and requires revision.

Line 244, the sentence ‘However, the rate of smoking among women has increased during the seven-year period’ requires revision. The increment was not much.

Line 308 the sentence requires revision. To include except ‘smoking cause heart attack’

Line 329, what omitted category refers to, reference category?.

Line 334-338, the statement quite confusing on ‘smoking causes serious illness based on the Table 4. Also the specific illness (stroke, heart attack, lung cancer, smokeless tobacco on serious illness) to be highlighted in the results section.

Line 380, the results on specific illnesses to be described.

The p value to be provided or denoted in the table(s) footnote.

For the tables, Exp B to be replaced with OR. For reference category, value 1 to be added.

Conclusion too long and to be incorporated into the discussion.

References to conform with the journal format.

6. PLOS authors have the option to publish the peer review history of their article (what does this mean?). If published, this will include your full peer review and any attached files.

Reviewer #1: **Yes: **Dharma N Bhatta

Reviewer #2: No

Reviewer #3: No

---

## [Author Response · Author response to Decision Letter 0]

22 Sep 2020

Response to Reviewers

The authors are grateful to the reviewers, who provided valuable comments and suggestions on earlier version of our work, which helped immensely in improving the quality of the paper.

Reviewer #1: 

1. Introduction section is lengthy, please concise it.

As per the suggestion we have concise the introduction part. 

2. Authors only focused on community or individual level determinant factor of smoking but tobacco industry activities, tobacco control policies and its implementation are also crucial on tobacco control.

The suggestion is valid however, no direct information is available in the Global Adult Tobacco Survey (GATS 1 and 2) dataset on tobacco industry activities, control policies and implementation. We have used decomposition analysis on both round the data set to capture these effect indirectly.

3. Authors provided several interventions during the study period from 2009to 2016 and the prevalence of tobacco use declined from 35% to 29%. If authors can provide previous than this study period declining trend on tobacco use in India, it is good to know the effects of these interventions. 

As per the suggestion we have added few lines in the introduction part related to trend.

4. I am unable to read reference no 15.

We have made the needful changes.

5. Last two paragraph of Introduction section needs to revise. Authors canprovide details of the analysis in supplemental file. Rationale is notmuch convincing why this study had done, why decomposition analysis,what are the difference compared to other analysis. If there any studyused other analysis approach and had limitations or any discrepancy.

As per the suggestion we have revised last two paragraph of Introduction section. Also revised the rationale of the study.

6. Who supported GATS in India, who was the implementation agency?

GATS 1 and 2, India is the project of the Ministry of Health & Family Welfare (MoHFW), Government of India. MoHFW designated the International Institute for Population Sciences (IIPS), Mumbai for GATS-1 and Tata Institute of Social Sciences (TISS), Mumbai for GATS-2 as the nodal implementing agency for the survey. We have included this information in the section ‘Description of dataset’. 

7. What does it mean by Indian residents? Citizen or any who reside inIndia for certain time.

In GATS survey the term India residents has been defined as those residents aged 15 or above, living in their usual residence prior to the survey date. The survey has excluded institutional population comprising those living in collective living places like students’ dormitories, hospitals, hotels, prisons, military barracks, etc

8. Sampling details is not available, how the household was selected? Howindividual was selected in the household?

As per the suggestion, we have included sampling details as follows

“The GATS is a nationally representative, multi-stage, geographically clustered sample of households that covered men and women above 15 years of age in India's 30 states (29 states in GATS-1) and two Union Territories (UTs). Multistage sampling procedure was adopted independently in each state, and within the states, independently in urban and rural areas to select the sample. In urban areas, a three stage sampling process was adopted. At the first stage, the list of all the wards from all cities and towns of the state/ UT constituted the urban sampling frame, from which a required sample of wards, i.e., primary sampling units (PSUs) was selected using probability proportional to size (PPS) sampling. At the second stage, a list of all census enumeration blocks (CEBs) in each selected ward constituted the sampling frame from which one CEB was selected by PPS from each ward. At the third stage, a list of all residential households in each selected CEB constituted the sampling frame, from which a sample of required number of households was selected. 

In rural areas, a two stage sampling process was adopted. At the first stage of sampling, PSUs (village) were selected using the PPS sampling method. At the second stage, a list of all residential households in each selected village constituted the sampling frame, from which a sample of the required number of households was selected. From each eligible household, one respondent was selected. More details about sampling design, training of the survey team, and survey management are separately documented in GATS-1 and GATS-2 published report.”

For ref., please see:

International Institute for Population Sciences (IIPS), Ministry of Health and Family Welfare (MoHFW), Government of India (2010) Global Adult Tobacco Survey India report (GATS India), 2009–10. New Delhi: MoHFW, Government of India; Mumbai: IIPS.

Tata Institute of Social Sciences (TISS), Ministry of Health and Family Welfare, Government of India. Global Adult Tobacco Survey GATS 2 India 2016-17 [Internet]. Available from: https://mohfw.gov.in/sites/default/files/GlobaltobacoJune2018.pdf.

9. What was the response rate for both period?

We have mentioned the response rate in the dataset description as follows .

“The overall response rate calculated as the product of the response rates at the household and person-level was 91.8 percent and 92.9 percent for GATS-1 and GATS-2 respectively.”

10. Please delete from Line 108 to 110. Just provide the references in thetext and authors need to provide details in the study.

We have deleted the above lines and provided the references in the Dataset description along with the details of both the surveys.

11. Line 112 to 115 is useless. In fact, the variables were not selected onthe basis of the literature, and the survey only included demographicvariables.

We have conducted a literature review mentioned in the second paragraph of the ‘Introduction section’ that led us to choose the selected variables. Further, the selection was based on the variables that would be associated with a decline in tobacco use and availability in both the rounds of the survey. 

12. Line 116 to 125 is not useful.

In line 116 to 125 we have mentioned about the recoding of states into six regions. In decomposition analysis result shows that regions are playing one of the important role in declining tobacco consumptions. Therefore, it is important to inform which states belongs to which region. 

13. I don't think caste and religion are household level category. Religionand caste should be an individual choice. What is backward classes andscheduled caste?

In Indian setting,caste and religion are considered as the household levelvariable as itdoes not vary from one household member to other household member. 

The “caste system” is an Indian social stratiﬁcation system which is used to represent the socio-economic status of an individual. The “caste” or “Jati” is hereditary and broadly divided into four groups for administrative reasons, namely, Scheduled Castes (SCs), Scheduled Tribes (STs), Other Backward Classes (OBCs), and General Castes (non-disadvantaged castes). For details, see Ref.,

The Pervasive and Persistent Influence of Caste on Child Mortality in India [Internet]. [cited 2020 Sep 3]. Available from: https://www.researchgate.net/publication/5153771_The_Pervasive_and_Persistent_Influence_of_Caste_on_Child_Mortality_in_India

14. How many household assets were included for wealth index? Should we include this in the methods?

For wealth index, 10 household assets were included in GATS 1 and 14 household assets were included in GATS 2. Based on time relevance, asset information has been captured in the surveys and that information has been used to create the wealth index in the respective time period. 

15. Line 140, what are the common variables? Please provide details aboutthe variables only which you had used in your study.

All the variables information has been shown in Table 1 separately for GATS-1 and 2 and variables not common have been marked with ‘NA’. In regression and decomposition analysis, we have explicitly usedthe common variables listedin the last part of the ‘Independent variables’ section(Line 173 to 175). 

16. It would be good to see any tobacco use in the dependent variable.

The objective of our paper is to decompose specifically the smoking and smokeless form of tobacco from GATS 1 to GATS 2. We have not included any tobacco use because it would not provide us a clear picture regarding which set of variables were responsible for a decline in smoking and smokeless tobacco usesaperatly. We have analysed the two forms of tobacco individually so that according to the results the government can plan an intervention or programme. 

17. What is the reference category for smoking only and smokeless only? Itincluded never user or former and never user.

We have revised the ‘Outcome variables’ section to specifically mention the reference category for smoking and smokeless includes those who are former and never user of smoking and smokeless tobacco.

18. How they calculated percent relative change? Was it a weighted percent?

We calculated relative change using the formula (Prevalence in GATS 2)-(Prevalence in GATS 1)/ Prevalence in GATS 1. We have use a weighted percent.

19. Line 155 to 173 is very hard to understand. It is better to specifydifferent analytic approaches separately. For example, why, when and howauthors used logistic (maybe) regression model? What they did with thisregression model? So on….. I don't think authors did multivariateanalysis.

First multivariate logistic regression models have been applied to find outcoefficientscorresponding to the different background characteristics. Thesecoefficientswere further used in the decomposition analysis.

We have applied separately four multivariate regression models: two regression models for smoking and smokeless tobacco in GATS 1 and two regression models for smoking and smokeless tobacco in GATS 2. 

As per the suggestion we have modified the ‘Statistical Analysis’ section to provide more clarity. 

20. It is better to provide details about decomposition analysis insupplemental file. Authors can provide step by step process of thisanalysis with SPSS syntax.

In SPSS, four separate logistic regression models have been applied to get the coefficients.Proportion has been calculated from GATS 1 and GATS 2 dataset. This proportion and rate we got from the logistic regression has been used todecomposethe tobacco decline analysis. 

21. Please specify, how the survey weights were used in the study?It is not clear whether authors combined both survey or analysedseparately, if combined how they treated survey weights.

Standard survey weights is given in the GATS 1 and GATS 2 file for population level estimate. For detailed information “The collected data was suitably weighted to improve representativeness of the sample in terms of size, distribution, and characteristics of the study population. The weights were derived considering design weight (reciprocal of the probability of selection), household response rate and individual response rate. Post-stratification calibration was done for ages-residence distribution on the survey period in each state/UT. Details of the weighting procedure are provided in Appendix B on Sample design in the GATS Report”.

We have analysed GATS 1 and GATS 2 separately and therefore there was no need to combine the survey weight. 

22. How the interaction worked in bivariate regression model? How propensitymatched in bivariate model? Which propensity match was used?

We have not included any interaction variable in the regression model. We got the Interaction term with the help of Decomposition analysis and the formula has been written in the paper. 

23. What was the accuracy of the decomposition model?

The decomposition technique adopted in this paper is a well-established demographic technique built upon Kitawaga’ (1955) classical work on rate standardization .

Kitagawa, E. M. (1955). Components of a difference between two rates. Journal of the American Statistical Association, 50, 1168–1194. https://doi.org/10.2307/2281213

24. Explain your model with Rate, Composition and Interaction.

According to the comments we have mentioned the explanation of Rate, Composition and Interaction in ‘Regression and Decomposition Analysis’ section.

25. Please specify proportion of the population in Tables 2 and 3. How itwas calculated?

Univariate analysis was performed for all background characteristics of smokers and smokeless tobacco users for both, GATS-1 and GATS-2. This analysis revealed a description of the respondents categorized in the twoforms of tobacco use, i.e. proportion of the population.

26. Line 201, …..and dual use tobacco…… I didn't see the dual usetobacco in tables and Methods, how dual use was defined? and why notpresented in the results?

We have removed the term ‘dual use’.

27. Very hard to follow Methods and Results section. Tables title showsmultivariate analysis but Methods section did not mention about it.

As per the suggestion we have added about multivariate analysis and try to simplify the language.

28. Interpretation of results is very hard to understand, please make itsimple and clear. Authors can focus on major significant results ontheir interpretation.

We have simplified the Results and Discussion section.

29. It would be better to put tables 2 and 3 in supplemental files. Inaddition to tables 4 and 5, I would suggest another table for AnyTobacco use.

As any tobacco is not our focus of interest, we have not included in the paper. Decomposition analysis is built upon proportion and rate, therefore we feel that table 2 and 3 should be in the main tables.

30. How Grand total was calculated in Tables 4 and 5. If the tables 4 and 5outcomes were from bivariate analysis, how could you eliminate thevariable selection bias?

While using the decomposition formula mentioned in the paper, first we got contribution of each and every attributes of listed variables and finaly we have added their contribution to get Grand total mentioned in Table 4 and 5. Table 4 and 5 were outcomes from multivariate logistic regression.

31. First paragraph of the Discussion section is repeated literature fromthe Introduction.

The first paragraph in the Discussion section enlists specific policy and programme interventions from the Government of India in the periodpreceding GATS 1.We have modified the Discussion section so that it comprehensively mentions all the programmes and interventions at one place in the Text.

32. Dual tobacco uses again came in the Discussion but there is no resultsabout dual use tobacco.

We have removed the term ‘dual use’.

33. Please provide the references for Line 461 to 464, please provide theseregions means which one…what interventions are available? Authorshould discuss potential contributing factors for higher and lowerregional reduction in tobacco use.

We have written this section again in line with the comments. We have discussed certain factors in the form of interventions and innovative programmes due to which certain regions have shown a higher contribution to the decline in tobacco use.

34. Discuss one factor at single place. For example, age is discussed inseveral paragraphs in the Discussion.

We have re-written the Discussion according to the feedback.

35. Conclusion is similar to discussion. Please rewrite your conclusion withyour study's main finding.

We have revised conclusion part of the paper.

36. Please try to include original articles in the references instead of internet websites. Some websites are not trusty.

We have updated the references according to the feedback.

Reviewer #2: This is a well written manuscript on comparing tobacco use

prevalence across two waves of Global Adult Tobacco Survey in India.

Authors have compared tobacco use prevalence data from GATS-1 and GATS-2

conduced in India to examine the demographic and socioeconomic

correlates ofsmoking and smokeless tobacco use for both the rounds of the survey.

Please see my feedback below.

General Comments

1. The manuscript may benefit from a good language and grammar editing.

We have tried to improve language and grammar part.

2. line 47-48: ADD Khaini under Smokeless tobacco (ST) use, which is themost prevalent form of ST use in India.

In line with the comments, we have added the term ‘Khaini’ as one of the prevalent forms of smokeless tobacco use in India.

3. Line 75-78: Please refer to these published papers on the theme:

Suliankatchi Abdulkader R, Sinha DN, Jeyashree K, et al. Trends intobacco consumption in India 1987-2016: impact of the World HealthOrganization Framework Convention on Tobacco Control. Into J PublicHealth. 2019;64(6):841-851. doi:10.1007/s00038-019-01252-x

Singh A, Arora M, Bentley R, et al Geographic variation in tobacco usein India: a population-based multilevel cross-sectional study BMJ Open2020;10:e033178. doi: 10.1136/bmjopen-2019-033178

We have gone through the above papers and it has helped us strengthen our Discussion section.

Material and Methods:

4. Role of Geographical areas need to be considered in this analysis,given published literature in using GATS has highlighted thatpeople's use of tobacco products varies by local areas (city ward andvillage) across India and the variation in this clustering by tobaccoproducts.

We have highlighted the role of Regions and used previous literature to supplement our findings in the Discussion section. However, absence of local areas like districts, city , ward and village in GATS survey, we are not able to explain variation within states.

5. Zones and Region has been used inter-changeably between text andTable1, thus confuses the reader.

We have corrected this and the term ‘Region’ is used throughout the text and Tables.

Discussion: Overall discussion needs to be revised to make it alignedwith results presented in the manuscript as it discusses and makesrecommendations on many issues not related to the results of thismanuscript.

We have revised discussion part .

6. Line 548-549: Pack warning in India as per rules and notification aresupposed to be in regional languages too. The rules mention that warninglanguage will be as per language used for branding. Please correct this.

We have corrected the statement.

7. Lines 554-555: Smokeless tobacco is also cheap as it is sold in smallsachets. how does affordability of single sale of cigarette compare tosale of bidis and smokeless tobacco? This explanation is not clear andnot aligned with results.

We have done correction in that line.

8. Lines 562-566: Unsure which result is being discussed here.

We have modified it to include the result from our decomposition analysis.

Overall check policy development years and details mentioned inIntroduction and discussion section of the manuscript.

We have rechecked the details and provided references for the same.

Reviewer #3: The manuscript entitled 'Declining Trend of Smoking andSmokeless Tobacco in India: A Decomposition Analysis' with the aim toexamine the socioeconomic correlates and delineate the factorscontributing to a change in smoking and smokeless tobacco.

The manuscript requires further improvement.

Comments

1. Statistical analysis

The description of the statistical analysis requires improvement and reorganization. A subtitle Statistical Analysis to be provided.Information on the variable coding to be provided. The word adjusted tobe used where applicable. The information on multicollinearity (if any),variable selection method in the analysis, interaction (if any),

goodness of fit/model fit etc to be stated.

According to the suggestion we have revised description of statistical analysis and reorganize it. We have added a section ‘Statistical analysis’ and reorganized it according to the feedback. We have provided variable coding , added adjusted word . We have mentioned variable selection method in the method section and checked multicollinearity and possibility of interaction term in the regression model. 

2. Line 150 -153, the coding for '0' categories to be provided.

We have provided the coding for “0” category.

3. Line 196-197, more information to be provided.

We have provided more information on the survey weight.

4. Line 197, typo 'Statistics version'.

We have corrected this error.

5. Results

Line 213, the results for dual use to be provided in the table(s).

We have removed the term ‘dual use’ from the Text.

6. Table 1, SCT, ST and OBC to be denoted in the table footnote. n to be

provided apart from %.

We have written the full description in the Methods section. 

7. Line 235, the sentence 'The Central region followed by the Western

region has attained a higher rate of smoking decline as opposed to other

regions' not clear based on the table and requires revision.

We have made the changes as follows “The eastern region followed by the central region has attained a higher rate of smoking decline as opposed to other regions”.

8. Line 244, the sentence 'However, the rate of smoking among women has

increased during the seven-year period' requires revision. The increment

was not much.

We have revised the line to read as “The result indicates that women have lower levels of smoking as compared to men in both the rounds of the survey”. 

9. Line 308 the sentence requires revision. To include except 'smokingcause heart attack'

We have revised the line according to the feedback “In the year 2016-17, the population who was unaware of ‘smoking as a cause of specific diseases excluding heart attack’ had higher levels of smokeless tobacco use as well”.

10. Line 329, what omitted category refers to, reference category?

Omitted category refers to reference category and we have made the changes in the manuscript.

11. Line 334-338, the statement quite confusing on 'smoking causes seriousillness based on the Table 4. Also the specific illness (stroke, heartattack, lung cancer, smokeless tobacco on serious illness) to behighlighted in the results section.

We have reworded the statement to make the impact of the knowledge factor more clear as follows “The negative sign in the sub-group who did not have knowledge of ‘smoking causing serious illnesses’ leads to the proposition that this group did not contribute to the decline in smoking with respect to the reference category. This implies that those having knowledge that smoking causes serious illnesses contributed more to lowering the rate of smoking”. 

We have highlighted the findings related to specific illnesses in the Results as follows “Those who were unaware that smoking causes stroke, heart attack and lung cancer contributed around 3%, 13% and 1% respectively while those unaware that smokeless tobacco causes serious illness contributed around 6% to the total decline in smoking”.

12. Line 380, the results on specific illnesses to be described.

We have described the results on specific illnesses as follows “Those who were aware that smoking causes serious illnesses, stroke, and lung cancer contributed to the decline in smokeless tobacco use. Those unaware that smoking causes heart attack contributed around four per cent to the total decline in smokeless tobacco use”. 

13. The p value to be provided or denoted in the table(s) footnote.

We have denoted the p value in the table footnote.

14. For the tables, Exp B to be replaced with OR. For reference category,

value 1 to be added.

We have replaced Exp B with AOR (adjusted odds ratio). We have added the value for the reference category in the tables and denoted it in the footnote.

15. Conclusion too long and to be incorporated into the discussion.

We have revised the Conclusion section according to the feedback.

16. References to conform with the journal format.

We have reworked the references.

---

## [Decision Letter · Decision Letter 1]

13 Oct 2020

PONE-D-20-16653R1

Declining Trend of Smoking and Smokeless Tobacco in India: A Decomposition Analysis

PLOS ONE

Dear Dr. Dixit,

Thank you for submitting your manuscript to PLOS ONE. After careful consideration, we feel that it has merit but does not fully meet PLOS ONE’s publication criteria as it currently stands. Therefore, we invite you to submit a revised version of the manuscript that addresses the points raised during the review process.

Reviewer 1 is not satisfied with your response and is now recommending rejection.  I am willing to give you one more chance to satisfy Reviewer 1.

The other reviewers are happy.

We look forward to receiving your revised manuscript.

Kind regards,

Stanton A. Glantz

Academic Editor

PLOS ONE

Reviewers' comments:

Reviewer's Responses to Questions

**Comments to the Author**

1. If the authors have adequately addressed your comments raised in a previous round of review and you feel that this manuscript is now acceptable for publication, you may indicate that here to bypass the “Comments to the Author” section, enter your conflict of interest statement in the “Confidential to Editor” section, and submit your "Accept" recommendation.

Reviewer #1: (No Response)

Reviewer #3: All comments have been addressed

2. Is the manuscript technically sound, and do the data support the conclusions?

Reviewer #1: Partly

Reviewer #3: Yes

3. Has the statistical analysis been performed appropriately and rigorously? 

Reviewer #1: No

Reviewer #3: Yes

4. Have the authors made all data underlying the findings in their manuscript fully available?

Reviewer #1: Yes

Reviewer #3: (No Response)

5. Is the manuscript presented in an intelligible fashion and written in standard English?

Reviewer #1: No

Reviewer #3: Yes

6. Review Comments to the Author

Reviewer #1: Authors have partially responded my previous comments and without complete corrections there are no further improvisations in the manuscript. I have major concern in their analysis process. This is a complex survey conducted in India and authors should follow the complex survey analyses on their analyses. I have commented about weights and weights related analyses in the entire manuscript. I have noted to provide brief analyses process of entire manuscript which could be published in supplemental file, especially how the numbers were calculated for Rate, Decomposition and Interaction in the tables. Authors did not provide explanations how their models were multivariate. Discussion and conclusion are very hard to follow.

Reviewer #3: (No Response)

7. PLOS authors have the option to publish the peer review history of their article (what does this mean?). If published, this will include your full peer review and any attached files.

Reviewer #1: **Yes: **Dharma N Bhatta

Reviewer #3: No

---

## [Author Response · Author response to Decision Letter 1]

26 Nov 2020

Reviewer #1: 

1. Authors have partially responded my previous comments and without complete corrections there are no further improvisations in the manuscript.

Explanation: As per reviewers suggestion we have revised the analysis, result, discussion and conclusion part of the paper.

2. I have major concern in their analysis process. This is a complex survey conducted in India and authors should follow the complex survey analyses on their analyses. 

Explanation: As per the reviewers valuable suggestion and in view of the complex survey, we have adopted the analysis accordingly. In the multivariate logistic regression analysis we have adjusted the impact of clustering and stratification. 

3. I have commented about weights and weights related analyses in the entire manuscript. 

Explanation: The analysis of the entire data including multivariate regression analysis has been carried out after assigning survey weights that is available in the GATS-1 and GATS-2 datasets. Further details of the weighting procedure are provided in Appendix B on Sample design in the GATS report (9,16). 

4. I have noted to provide brief analyses process of entire manuscript which could be published in supplemental file, especially how the numbers were calculated for Rate, Decomposition and Interaction in the tables. 

Explanation: As per reviewers suggestion we have provided syntax in the supplementary file and added an explanation . 

5. Authors did not provide explanations how their models were multivariate. Discussion and conclusion are very hard to follow.

Explanation: We have applied Multivariate binary logistic regression model to investigate the adjusted associations of socioeconomic, demographic and knowledge correlates of tobacco consumption in the smoking and smokeless forms in India. In this paper we have applied four multivariate logistic regression models. In all these regression models we have included 18 covariates together to get an adjusted estimates from multivariate model. We have also provided syntax in supplementary file.

As per the suggestion we have revised the discussion and conclusion part.

---

## [Decision Letter · Decision Letter 2]

7 Dec 2020

PONE-D-20-16653R2

Declining Trend of Smoking and Smokeless Tobacco in India: A Decomposition Analysis

PLOS ONE

Dear Dr. Dixit,

Thank you for submitting your manuscript to PLOS ONE. After careful consideration, we feel that it has merit but does not fully meet PLOS ONE’s publication criteria as it currently stands. Therefore, we invite you to submit a revised version of the manuscript that addresses the points raised during the review process.

We look forward to receiving your revised manuscript.

Kind regards,

Stanton A. Glantz

Academic Editor

PLOS ONE

Reviewers' comments:

Reviewer's Responses to Questions

**Comments to the Author**

1. If the authors have adequately addressed your comments raised in a previous round of review and you feel that this manuscript is now acceptable for publication, you may indicate that here to bypass the “Comments to the Author” section, enter your conflict of interest statement in the “Confidential to Editor” section, and submit your "Accept" recommendation.

Reviewer #1: (No Response)

2. Is the manuscript technically sound, and do the data support the conclusions?

Reviewer #1: (No Response)

3. Has the statistical analysis been performed appropriately and rigorously? 

Reviewer #1: (No Response)

4. Have the authors made all data underlying the findings in their manuscript fully available?

Reviewer #1: (No Response)

5. Is the manuscript presented in an intelligible fashion and written in standard English?

Reviewer #1: (No Response)

6. Review Comments to the Author

Reviewer #1: How authors applied data weights in their analysis is still not clear. Details are not available in the manuscript. Authors referred reference 9 and 16 Appendix B for weight calculation procedures, but referred document did not provide details about how to use calculated weights. Authors should provide detailed procedures how they applied calculated weights in their entire outcomes. Were they use STRATA, PSU and Weights or STRATA and Weights or PSU and Weights or only Weights? I don’t see any weight related variables in the syntax as well. I guess, there are different steps in SPSS than other statistical software for complex sample analysis. How they develop .csplan file in SPSS for complex sample or How they prepared “Analysis Preparation Wizard”? Were these all available GATS dataset.

After reading their syntax, I don’t think their logistic regression is multivariate, it should be multivariable logistic regression.

For decomposition, detailed provided in the supplemental file should be in the Method section instead of that complex equation.

4th para of conclusion, limitations of the paper should be incorporated in the Discussion section.

There are redundancies in the Conclusions section and its longer than Discussion. Conclusions should be within one short para probably 4 to 5 sentences.

7. PLOS authors have the option to publish the peer review history of their article (what does this mean?). If published, this will include your full peer review and any attached files.

Reviewer #1: **Yes: **Dharma N Bhatta

---

## [Author Response · Author response to Decision Letter 2]

28 Dec 2020

Reviewers Response

Reviewer #1: How authors applied data weights in their analysis is still not clear. Details are not available in the manuscript. Authors referred reference 9 and 16 Appendix B for weight calculation procedures, but referred document did not provide details about how to use calculated weights. Authors should provide detailed procedures how they applied calculated weights in their entire outcomes. Were they use STRATA, PSU and Weights or STRATA and Weights or PSU and Weights or only Weights? I don’t see any weight related variables in the syntax as well. 

Explanation: Survey weight is given in the GATS-1 and 2 data files. Initially we have performed analysis in SPSS software and used syntax weight on. After reviewers suggestion, to adjust complex analysis we have adjusted clustering and stratum effect while doing analysis in STATA-14 software and applied the survey weight to handle the complex survey design. 

While generating all the tables of this paper, each record (individual case) was multiplied by survey weight. These weights were estimated for adjustment of 1) unequal probability of selection, 2) differential response rates across states and male/ female in rural/ urban areas within the states and 3) differences in the distribution of survey population and actual population (projected as on survey period) of each state by rural/urban areas and by sex and broad age-group.

In other words, the weights were the adjustment within each individual state and across the states. 

Further details of the weighting procedure are provided in section A 4 page 216, on GATS-1 report.

https://www.healis.org/pdf/special-report/GATS_1.pdf

2. I guess, there are different steps in SPSS than other statistical software for complex sample analysis. How they develop .csplan file in SPSS for complex sample or How they prepared “Analysis Preparation Wizard”? Were these all available GATS dataset.

Explanation: Complex analysis adjustment has been done in stata software. We have used the command svyset gatscluster [pweight= gatsweight], strata (gatsstrata). All the mentioned variable are available in GATS dataset.

3. After reading their syntax, I don’t think their logistic regression is multivariate, it should be multivariable logistic regression.

Explanation : The term multivariate and multivariable are often used interchangeably in the public health literature. However, as per the suggestion we have used multivariable logistic regression in manuscript.

4. For decomposition, detailed provided in the supplemental file should be in the Method section instead of that complex equation.

Explanation : As per the journal guideline we have written question in the method section and according to readers ease of understanding question explanation has been given in supplementary file.

5. 4th para of conclusion, limitations of the paper should be incorporated in the Discussion section. There are redundancies in the Conclusions section and its longer than Discussion. Conclusions should be within one short para probably 4 to 5 sentences.

Explanation :As per the suggestion we have revised the conclusion part and added limitation in the discussion part of the paper.

---

## [Decision Letter · Decision Letter 3]

25 Jan 2021

PONE-D-20-16653R3

Declining Trend of Smoking and Smokeless Tobacco in India: A Decomposition Analysis

PLOS ONE

Dear Dr. Dixit,

Thank you for submitting your manuscript to PLOS ONE. After careful consideration, we feel that it has merit but does not fully meet PLOS ONE’s publication criteria as it currently stands. Therefore, we invite you to submit a revised version of the manuscript that addresses the points raised during the review process.

Please add the methodological details and make the other clarifications that the reviewer suggested to the manuscript.

We look forward to receiving your revised manuscript.

Kind regards,

Stanton A. Glantz

Academic Editor

PLOS ONE

Reviewers' comments:

Reviewer's Responses to Questions

**Comments to the Author**

1. If the authors have adequately addressed your comments raised in a previous round of review and you feel that this manuscript is now acceptable for publication, you may indicate that here to bypass the “Comments to the Author” section, enter your conflict of interest statement in the “Confidential to Editor” section, and submit your "Accept" recommendation.

Reviewer #1: (No Response)

2. Is the manuscript technically sound, and do the data support the conclusions?

Reviewer #1: (No Response)

3. Has the statistical analysis been performed appropriately and rigorously? 

Reviewer #1: (No Response)

4. Have the authors made all data underlying the findings in their manuscript fully available?

Reviewer #1: (No Response)

5. Is the manuscript presented in an intelligible fashion and written in standard English?

Reviewer #1: (No Response)

6. Review Comments to the Author

Reviewer #1: Authors provided following responses to my weight related comments in the response section only but did not included in the manuscript. I would like to suggest authors to include these explanations into the Analysis section of the manuscript, so that readers can understand how authors had done their analysis.

(Comment 1): Explanation: Survey weight is given in the GATS-1 and 2 data files. Initially we have

performed analysis in SPSS software and used syntax weight on. After reviewers suggestion,

to adjust complex analysis we have adjusted clustering and stratum effect while doing analysis

in STATA-14 software and applied the survey weight to handle the complex survey design.

While generating all the tables of this paper, each record (individual case) was multiplied by

survey weight. These weights were estimated for adjustment of 1) unequal probability of

selection, 2) differential response rates across states and male/ female in rural/ urban areas

within the states and 3) differences in the distribution of survey population and actual

population (projected as on survey period) of each state by rural/urban areas and by sex and

broad age-group.

In other words, the weights were the adjustment within each individual state and across the

states.

Further details of the weighting procedure are provided in section A 4 page 216, on GATS-1

report.

https://www.healis.org/pdf/special-report/GATS_1.pdf

(Comment 2): Explanation: Complex analysis adjustment has been done in stata software. We have used the

command svyset gatscluster [pweight= gatsweight], strata (gatsstrata). All the

mentioned variable are available in GATS dataset.

7. PLOS authors have the option to publish the peer review history of their article (what does this mean?). If published, this will include your full peer review and any attached files.

Reviewer #1: **Yes: **Dharma N Bhatta

---

## [Author Response · Author response to Decision Letter 3]

1 Feb 2021

Reviewer #1: Authors provided following responses to my weight related comments in the response section only but did not included in the manuscript. I would like to suggest authors to include these explanations into the Analysis section of the manuscript, so that readers can understand how authors had done their analysis.

Explanation: As per suggestion we have included the explanation in the analysis section of the paper.

---

## [Editor Report · Decision Letter 4]

4 Feb 2021

Declining Trend of Smoking and Smokeless Tobacco in India: A Decomposition Analysis

PONE-D-20-16653R4

Dear Dr. Dixit,

We’re pleased to inform you that your manuscript has been judged scientifically suitable for publication and will be formally accepted for publication once it meets all outstanding technical requirements.

Kind regards,

Stanton A. Glantz

Academic Editor

PLOS ONE
---

## [Editor Report · Acceptance letter]

8 Feb 2021

PONE-D-20-16653R4 

Declining Trend of Smoking and Smokeless Tobacco in India: A Decomposition Analysis 

Dear Dr. Dixit:

I'm pleased to inform you that your manuscript has been deemed suitable for publication in PLOS ONE. Congratulations! Your manuscript is now with our production department. 

Kind regards, 

on behalf of

Professor Stanton A. Glantz 

Academic Editor

PLOS ONE